# Reinforcement learning with combinatorial actions for coupled restless bandits

**Lily Xu,**[1,2][*] **Bryan Wilder,**[3] **Elias B. Khalil,**[4] **Milind Tambe**[1]
[1]Harvard University, [2]University of Oxford, [3]Carnegie Mellon University, [4]University of Toronto

## Abstract

Reinforcement learning (RL) has increasingly been applied to solve real-world planning problems, with progress in handling large state spaces and time horizons. However, a key bottleneck in many domains is that RL methods cannot accommodate large, combinatorially structured action spaces. In such settings, even representing the set of feasible actions at a single step may require a complex discrete optimization formulation. We leverage recent advances in embedding trained neural networks into optimization problems to propose SEQUOIA, an RL algorithm that directly optimizes for long-term reward over the feasible action space. Our approach embeds a Q-network into a mixed-integer program to select a combinatorial action in each timestep. Here, we focus on planning over restless bandits, a class of planning problems which capture many real-world examples of sequential decision making. We introduce CORMAB, a broader class of restless bandits with combinatorial actions that cannot be decoupled across the arms of the restless bandit, requiring direct solving over the joint, exponentially large action space. We empirically validate SEQUOIA on four novel restless bandit problems with combinatorial constraints: multiple interventions, path constraints, bipartite matching, and capacity constraints. Our approach significantly outperforms existing methods—which cannot address sequential planning and combinatorial selection simultaneously—by an average of 24.8% on these difficult instances.

## 1 Introduction

Reinforcement learning (RL) has made tremendous progress in recent years to solve a wide range of practical problems (Treloar et al., 2020; Marot et al., 2021; Silvestro et al., 2022; Degrave et al., 2022). While successful at dealing with large or infinite state spaces, RL struggles with discrete, combinatorial action spaces. This limitation is pertinent to many real-world sequential decision-making problems, where resource constraints frequently lead to combinatorial action spaces (Dulac-Arnold et al., 2020). Consider, for example, a sequential resource allocation problem in which public health workers are dispatched to visit patients. The workers each have a limited daily budget to maximize patient well-being. These requirements give rise to an exponentially large combinatorial action space to optimize over, even when the number of workers and patients is small.

A suitable framework for modeling such problems is that of *restless multi-armed bandits* (Niño-Mora, 2023), which have been applied to settings from clinical trial design (Villar et al., 2015) to behavioral interventions (Mate et al., 2022). An "arm" of the restless bandit corresponds to a patient in the aforementioned public health example. At every timestep, each arm is in one of a finite number of states (e.g., representing the health of that individual), and we aim to select actions so that arms move into the more beneficial states. Typically, restless bandit solutions have been confined to relatively simple and thus tractable problem structures. For example, the common Whittle index policy (Whittle, 1988) requires that each arm be independent, each action impacts only a single arm, and arms transition independently (Figure 1A). Under these assumptions, the problem admits a threshold-based top-$K$ policy which decouples the combinatorial action selection by learning policies for each arm independently.

Real-world applications, on the other hand, often involve complicated problem structures beyond the simple budget constraint, for which action decomposition would substantially degrade performance.

---

[*]Contact: `lily.x@columbia.edu`. Code available at: https://github.com/lily-x/combinatorial-rmab

Figure 1: Restless bandits are a variant of planning over Markov decision processes, where each "arm" transitions depending on whether it is acted upon. Standard restless bandits can be solved with threshold-based policies, but these approaches are unable to address challenging settings with combinatorial constraints on the actions that cannot be decoupled, prohibiting the application of easy heuristic solutions. Our paper considers this class of strongly coupled restless bandits (CORMAB). We describe these new problem formulations (**A–D**) for restless bandits in detail in Section 3.

Figure 1A–D illustrate four such scenarios. For example, consider the problem of planning a route for a health worker to visit patients within a travel constraint (see Figure 1D): each action (path) impacts multiple patients and cannot be decomposed further (e.g., into separate edges) while ensuring the overall path is feasible. Another example is where actions represent heterogeneous interventions that may impact not just a single arm but several simultaneously. Individuals may be acted on by multiple actions simultaneously (e.g., exposure to different messaging campaigns or nurse visits), with a potentially nonlinear cumulative effect (Figure 1B).

In these more complex settings, *we can no longer decompose the actions* and must reason about the entire combinatorial structure at once. Coupled with combinatorial constraints, the discrete action space becomes notoriously difficult to optimize the reward function over. Thus, existing algorithms cannot accommodate this class of problems. To that end, we consider the restless bandit problem with combinatorial actions (which we call CORMAB) in four problem settings (Figure 1A–D, described in Section 3) that highlight its broad applicability.

The existing literature at the intersection of RL and combinatorial optimization has not considered sequential problems where the action space in each step is combinatorial. Rather, existing approaches solve *single-stage* combinatorial optimization problems with RL by decomposing them into iterative choices from a small action set, such as iteratively choosing the next node in a traveling salesperson or vehicle routing problem (Dai et al., 2017; Barrett et al., 2020; Delarue et al., 2020).

Here, we consider for the first time *sequential* combinatorial settings where the reward comes not from a single-shot action but is incurred after enacting a policy over many timesteps. We leverage recent advances in integrating deep learning with mathematical programming, in which a neural network with ReLU activations can be directly embedded into a mixed-integer program (Fischetti & Jo, 2018; Serra et al., 2018). By integrating this technique with an RL training procedure, we show how to learn the long-term reward using a deep Q-network to approximate the value function, while tractably optimizing this Q-network over the discrete combinatorial space at each step with a mixed-integer program. We call our algorithm SEQUOIA, for SEQUential cOmbInatorial Actions.

Our paper contributes: (1) four new problem formulations for restless bandits with combinatorial action constraints, (2) a general-purpose solution approach that combines deep Q-learning with mathematical programming, and (3) empirical evaluations demonstrating the strength of this approach. Our work opens up a broad class of problem settings for restless bandits with more complex action constraints. Beyond restless bandits, our work enables general RL planning over Markov decision processes with per-timestep combinatorial action spaces.

## 2   PROBLEM DESCRIPTION

We consider offline, stochastic planning for restless bandits with combinatorial action constraints, where the transition dynamics and reward are known a priori, but the state and action space are too large to be solved directly. Specifically, we consider the setting where the arms are *strongly coupled* so we cannot decouple the arms of the bandit into independently controlled Markov chains using a Lagrangian relaxation. We begin by describing the standard restless bandit, then present our general formulation for *constrained combinatorial-action RMABs*, which we refer to as CORMAB.

## 2.1 STANDARD RESTLESS BANDITS

The standard restless bandit problem is modeled by a set of $J$ Markov decision processes (MDPs). Each MDP $(\mathcal{S}, \mathcal{A}, \mathcal{P}, R)$ represents one arm of the restless bandit. The state $s_j \in \mathcal{S}$ of an arm $j$ transitions to next state $s_j'$ depending on the selected action $a_j \in \mathcal{A}$, with known probability $P_j(s_j, a_j, s_j') : \mathcal{S} \times \mathcal{A} \times \mathcal{S} \to [0, 1]$. We use the vector $\boldsymbol{s} \in \mathcal{S}^\times$ to denote the joint state of all $J$ arms, and similarly $\boldsymbol{a} \in \mathcal{A}^\times$ for the joint action. Traditional restless bandits assume that each action acts on a single arm only, enabling an optimal policy that *decouples* the solution into independent, per-arm subproblems.

Each state $s_j \in \mathcal{S}$ has an associated reward $r_j(s_j)$ and the total reward at each timestep $t$ is the sum of the rewards of the arms: $R^{(t)}(\boldsymbol{s}) = \sum_{j=1}^{J} r_j(s_j)$. The traditional restless bandit problem requires selecting a binary action $a_j \in \{0, 1\}$ for each arm, subject to a total budget constraint $\sum_{j=1}^{J} a_j \leq B$.

## 2.2 RESTLESS BANDITS WITH STRONGLY COUPLED ACTIONS (CORMAB)

We now introduce CORMABs, a broader class of problems for restless bandits where actions may be *strongly coupled* due to combinatorial constraints. We consider a set of $N$ actions $a_i$ that each impact the transition probability of some subset of the arms. At each timestep, we must pick a *combinatorial* action over the arms i.e., the action vector $\boldsymbol{a}$ must be in $\mathcal{C} \subseteq \mathcal{A}^\times$, a set defined by constraints. (We provide example instantiations of $\mathcal{C}$ in Section 3.) Now, the transition probabilities may be coupled, with individual actions impacting multiple arms. We denote the joint transition probability as $P^\times : \mathcal{S}^J \times \mathcal{A}^N \times \mathcal{S}^J \to [0, 1]^J$.

From a given state $\boldsymbol{s}$, by Bellman's optimality principle we seek the combinatorial action $\boldsymbol{a}$ that maximizes the expected long-term reward, with discount factor $\gamma \in (0, 1]$:

$$\max_{\boldsymbol{a} \in \mathcal{C}} Q(\boldsymbol{s}, \boldsymbol{a}) \tag{1}$$

$$\text{s.t. } Q(\boldsymbol{s}, \boldsymbol{a}) = R(\boldsymbol{s}) + \gamma \sum_{\boldsymbol{s}' \in \mathcal{S}} P^\times(\boldsymbol{s}, \boldsymbol{a}, \boldsymbol{s}') V(\boldsymbol{s}') \qquad \forall \boldsymbol{s} \in \mathcal{S}^\times, \boldsymbol{a} \in \mathcal{C}$$

$$V(\boldsymbol{s}) = \max_{\boldsymbol{a} \in \mathcal{C}} Q(\boldsymbol{s}, \boldsymbol{a}) \qquad \forall \boldsymbol{s} \in \mathcal{S}^\times$$

This value function is difficult to implement as the $\max$ over actions $\boldsymbol{a}$ is taken over a combinatorial number of possible actions, and there are a combinatorial number of constraints (for each possible state $\boldsymbol{s}$). Enumerative methods are therefore intractable. We address CORMAB problems of this form using a combination of RL to learn the $Q$-function and mixed-integer programming to find an optimal constraint-feasible action $\boldsymbol{a}$.

## 3 ENABLING NEW PROBLEM FORMULATIONS FOR RESTLESS BANDITS

We introduce four instantiations of CORMAB problems which cannot be modeled or solved by standard restless bandit approaches. These formulations consider compounding effects from multiple interventions, path planning, bipartite matching, and capacity constraints—each of which is a widely applicable challenge for resource allocation problems. To the best of our knowledge, the following problem formulations are all novel for restless bandits.

In each case, action selection requires various forms of constrained optimization. Note that planning in each instance still requires specifying the $Q$-function, which serves as the objective in each setting; in Section 4, we will address how to resolve estimating the $Q$-function. Additional details, including MILP formulations for each problem setting, are provided in Appendix C.

**Multiple interventions for public health** In public health programs, different informational campaigns may impact different groups of individuals. Interventions could include, for example, radio, ads at bus stations, church events, or fliers. We model this problem as a restless bandit where one action can impact a fixed subset of arms. We propose the first combinatorial setting in which (a) one action impacts multiple arms, and (b) multiple actions may be applied simultaneously to each arm with compounding effects. One example of this setting is visualized in Figure 1A.

Here, each action $a_i \in \{0, 1\}$ corresponds to one of $N$ possible messaging interventions, each arm $j$ corresponds to a patient, and the state $s_j$ of a patient represents their engagement level in a health program. In this model, we consider $s_j \in \{0, 1, 2, 3\}$ where each is associated with a higher level of engagement. More details in Appendix B.2. The transition probability for patient $j$ in state $s_j$ transitioning to a higher $s_j^+$ or lower $s_j^-$ state is:

$$P_j(s_j, \boldsymbol{a}, s_j^+) = \phi\left(\omega_j(s_j)^\top \boldsymbol{a}\right), \qquad P_j(s_j, \boldsymbol{a}, s_j^-) = 1 - P_j(s_j, \boldsymbol{a}, s_j^+), \qquad (2)$$

where $\omega_j(s_j) \in \mathbb{R}^N$ is an individual and state-specific coefficient vector describing their response to different interventions, $\boldsymbol{a}$ is the action vector, and $\phi : \mathbb{R} \to [0, 1]$ is a known link function. Each action has a cost $c_i$ and we are limited by a total budget $B$, giving rise to the optimization problem:

$$\max_{\boldsymbol{a}} \; Q(\boldsymbol{s}, \boldsymbol{a}) \;\; \text{s.t.} \sum_{j \in [J]} c_j a_j \leq B; \quad a_j \in \{0, 1\} \quad \forall j \in [J].$$

Note that this combinatorial problem formulation generalizes the standard restless bandit: we can recover the original restless bandit setting by considering $N = J$ actions, where the $j$th action is connected to the $j$th arm and the edge weight $\omega_j$ equals the transition probability $P_j(s_j, a_j = 1, s_j')$. In our experiments, we consider a sigmoid link function, which has been widely used in the public health literature (Levy et al., 2006). We show that using a piecewise linear approximation of this link function, which can be easily encoded into the MILP solver, solves the problem effectively.

**Schedule-constrained CORMAB (Bipartite matching)**  Here, actions (e.g., volunteer health workers) and arms (e.g., patients) both have limited time windows for scheduling, with $K$ available timeslots. We must assign workers to patients to form a valid schedule, where workers and patients are matched such that they have mutually agreed upon times. Each worker can only be assigned once. This problem resembles bipartite matching (Figure 1B).

**Capacity-constrained CORMAB**  Suppose we have $N$ workers each with a budget constraint $b_i$, and the $J$ arms each have some cost $c_j$ for contacting them (Figure 1C). This problem is an instance of the NP-hard generalized assignment problem (Özbakir et al., 2010). The goal is to assign workers to patients while respecting each worker's capacity constraint.

**Path-constrained CORMAB**  Now suppose that each arm of the restless bandit lies in a network, represented as a graph $(\mathcal{V}, \mathcal{E})$ (see Figure 1D). For example, nodes may correspond to patients' residential locations if we are scheduling at-home patient interventions (Kergosien et al., 2009). We have one arm at each node $j \in \mathcal{V}$, with edges $(j, k) \in \mathcal{E}$ connecting the nodes. We include self-loops, so $(j, j) \in \mathcal{E}$ for all $j \in \mathcal{V}$. The action is to pick a path along the edges in $\mathcal{E}$ with a constraint on its total length $T$; we act on each arm whose node we visit along the path.

## 4  SOLVING RMABS WITH COMBINATORIAL ACTIONS

We present a novel algorithm, SEQUOIA, which builds on deep Q-learning (DQN) by integrating a mixed-integer linear program (MILP) for combinatorial action selection. We start by introducing the basic DQN+MILP framework in Section 4.1. In Section 4.2, we provide a set of strategies to find a good initialization for the Q-network and improve computational efficiency. We visualize our approach in Figure 2, and Algorithm 1 provides pseudocode for the Q-network training procedure.

While we focus on the restless bandit problem setting, SEQUOIA generalizes to other sequential planning problems with per-timestep combinatorial actions, provided that the restrictions on the actions can be represented as constraints in a mixed-integer program.

### 4.1  Q-LEARNING WITH COMBINATORIAL ACTIONS

Deep Q-learning aims to estimate the long-run value of action $\boldsymbol{a}$ from state $\boldsymbol{s}$, denoted $Q(\boldsymbol{s}, \boldsymbol{a})$, by parameterizing the Q-function with a neural network. The DQN algorithm (Mnih, 2013) samples *(action, next state, reward)* sequences via a mix of random actions and on-policy samples (i.e., actions that maximize the current estimate of $Q$). The samples are used to improve the Q-function via temporal difference (TD) updates which aim to minimize the residual in the Bellman equation.

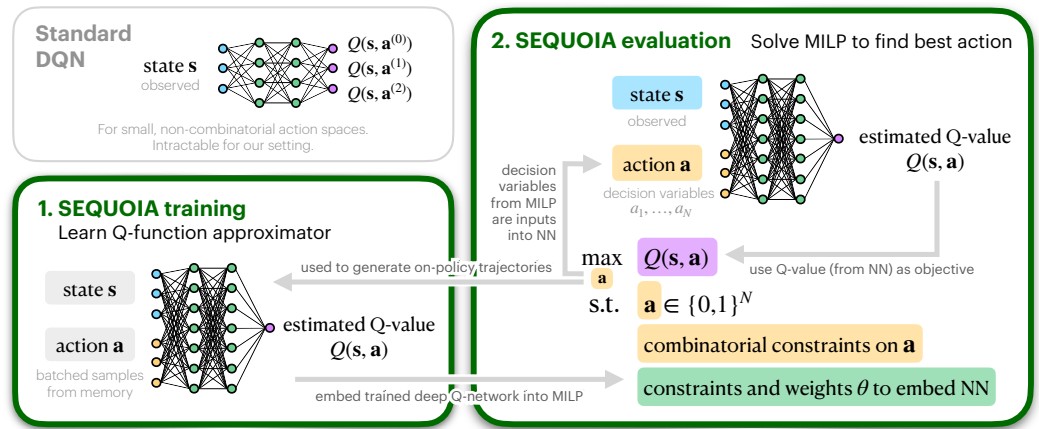

Figure 2: An overview of our SEQUOIA algorithm. Standard DQN takes as input the state and outputs estimated Q-values for actions, which are assumed to be easily enumerable. In contrast, we consider cases when the actions are too large to be enumerated due to their combinatorial constraint structure. **PART 1.** We therefore train a Q-network where the action $a$ is included as an input (Algorithm 1; described in Section 4.1). **PART 2.** We then embed that Q-network into a mixed-integer program, which also specifies the combinatorial action constraints (e.g., the formulations provided in Section 3). In evaluating the objective, *the MILP solver conducts a forward pass through the neural network* in order to calculate the expected Q-value. Solving the MILP thus finds an action $a$ (the decision variables) that maximizes the predicted Q-value $Q(s, a)$ (our objective).

This framework breaks down when the action space is exponentially large, as in our setting. The first challenge is that typical uses of DQN output Q-values for all actions simultaneously by mapping the state to one network output per action. Since our domain has an exponentially large action space, we instead use DQN to estimate the Q-value of a given state–action *pair*, where the action is represented as a binary vector $a$ of length $N$ (Figure 2, Part 1).

While this allows for a concise representation of the Q-function, it leaves a critical bottleneck: computing the temporal difference loss requires identifying the best action *on policy*, that is, according to the current Q-function (Algorithm 1, lines 9 and 13). That is, it requires solving problems of the form $\max_a Q(s, a; \theta)$ where $\theta$ denotes the parameters of the Q-network. With standard DQN, this problem would be solved by evaluating all actions and picking the best one, but brute-force enumeration is infeasible for combinatorial spaces.

Instead, we leverage recent advances that enable encoding a neural network into a mixed-integer linear program to enable optimization over the trained network (Fischetti & Jo, 2018; Serra et al., 2018). Using these methods, we embed an extra set of variables and constraints in the MILP. This is possible because neural networks are specified as a series of linear inequalities, commonly connected with a rectified linear unit (ReLU) as the activation function. Such a neural network can thus be modeled as a binary mixed-integer linear program, with a continuous variable representing the output of each hidden unit and a binary variable representing the output of each ReLU. Specifically, a single hidden unit in a neural network is represented as:

$$x = \sigma(w^\top y + b) \tag{3}$$

where the activation function $\sigma$ is the MILP-representable ReLU: $\sigma(a) = \text{ReLU}(a) = \max\{0, a\}$. Fischetti & Jo (2018) show that this hidden unit can be expressed as:

$$w^\top y + b = x - s, \quad x \geq 0, \quad s \geq 0 \tag{4}$$
$$z = 1 \implies x \leq 0, \quad z = 0 \implies s \leq 0 \tag{5}$$
$$z \in \{0, 1\} \tag{6}$$

with binary variable $z$ and continuous variables $x$ and $s$. The two logical constraints (eq. 5) are translated into big-M constraints. In this way, a fully-connected feed-forward neural network with $D$ hidden layers and $P$ neurons per layer can be represented exactly within a MILP with $\mathcal{O}(DP)$ binary variables and linear constraints (see Section 4.2.1 of Huchette et al. (2023)).

---

**Algorithm 1** SEQUOIA for RL with combinatorial actions

---

**Input**: MDP instance $(\mathcal{S}, \mathcal{A}, \mathcal{P}, R)$ and a set of constraints $\mathcal{C} \subseteq \mathcal{A}$
**Parameter**: Epsilon-greedy parameter $\epsilon$, target update frequency $F$

1: Initialize action–value function $Q$ with weights $\theta$
2: Strategically generate actions, and store state–action samples in memory $\mathcal{D}$
3: Pre-train $Q$ with myopic observed reward, using state–action samples from $\mathcal{D}$
4: Initialize target network $\hat{Q}$: $\theta^- = \theta$
5: Construct MILP with current network weights $\theta$ and constraints $\mathcal{C}$
6: **for** episode $= 1, \ldots, E$ **do**
7:    **for** $t = 1, \ldots, T$ **do**
8:       With probability $\epsilon$, select random action $\boldsymbol{a}^{(t)} \in \mathcal{C}$
9:       Otherwise, select $\boldsymbol{a}^{(t)} = \arg\max_{\boldsymbol{a}} \mathrm{MILP}(\boldsymbol{s}^{(t)}; \theta^-)$
10:      Execute action $\boldsymbol{a}^{(t)}$, calculate expected reward $R^{(t)}$, and observe next state $\boldsymbol{s}^{(t+1)}$
11:      Store transition $(\boldsymbol{s}^{(t)}, \boldsymbol{a}^{(t)}, R^{(t)}, \boldsymbol{s}^{(t+1)})$ in $\mathcal{D}$
12:      Sample random minibatch of transitions $(\boldsymbol{s}^{(k)}, \boldsymbol{a}^{(k)}, R^{(k)}, \boldsymbol{s}^{(k+1)})$ from $\mathcal{D}$
13:      Set $y^{(k)} = R^{(k)} + \gamma \max_{\boldsymbol{a}} \mathrm{MILP}(\boldsymbol{s}^{(k)}; \theta^-)$
14:      Gradient descent step on $(y^{(k)} - Q(\boldsymbol{s}^{(k)}, \boldsymbol{a}^{(k)}; \theta))^2$
15:      Every $F$ steps, update $\hat{Q} = Q$
16: **return** Q-function $Q$

---

Specifically, we use the MILP to represent the Q-function $Q(\boldsymbol{s}, \boldsymbol{a})$, the objective of our planning problem, as specified in the problem formulations in Section 3. We therefore embed the trained Q-network into the MILP formulations of our constraints and use its output as our objective. Thus every time we solve the MILP to determine an optimal action, we implicitly simulate a forward pass of the neural network to calculate the Q-value. In every new episode, we build a new MILP instance which includes constraints that represent the current policy network $Q$ with parameters $\theta$, along with the original constraints on the actions. Then the inner step of $\arg\max_{\boldsymbol{a}}$ (line 9) solves that MILP to pick a combinatorial action $\boldsymbol{a}$ for state $\boldsymbol{s}$ that maximizes the expected long-run return of pair $(\boldsymbol{s}, \boldsymbol{a})$, based on estimates from the neural network $Q$ (Figure 2, Part 2).

Typically, DQN requires executing the current policy as determined by the current Q-network to get on-policy samples. However, whereas standard DQN assumes we can enumerate all actions to pick the one with the highest $Q$-value, doing so is intractable in our setting (Figure 2). Instead, we must solve the MILP, using the current network parameters $\theta^-$, in order to determine an action. To both encourage exploration and reduce the computational complexity, we select a random feasible action with probability $\epsilon$ (line 8).

Our algorithm runs this modified DQN training procedure. On top of these key changes to accommodate combinatorial actions, we incorporate standard best practices for DQN (Hessel et al., 2018), including double DQN (Van Hasselt et al., 2016), prioritized experience replay (Schaul et al., 2016), and gradient clipping (Zhang et al., 2020).

Once the Q-network is fully trained, we use this network for planning. At inference time, we solve the MILP once at every timestep, computing the best action to take from current state $\boldsymbol{s}$ using the MILP encoding of the final Q-network (with parameters $\theta$) and the action constraints. In other words, we solve for: $\max_{\boldsymbol{a}} \mathrm{MILP}(\boldsymbol{s}; \theta)$, depicted in Figure 2, Part 2.

## 4.2 SMART ACTION GENERATION AND OTHER COMPUTATIONAL ENHANCEMENTS

There are two independent critical computational bottlenecks with integrating neural networks into MILPs. First, solving the MILP at every timestep is necessary for both training and evaluation, but is extremely expensive: for the discounted future reward estimate in line 13, we must perform that calculation for *every sample in the minibatch* at every timestep. For a training instance with $E$ episodes, time horizon $H$, and minibatches of size $M$, we have $EHM$ repeated MILP solves. Even a modest problem setting of $E = 1,000$, $H = 20$, and $M = 32$ requires a prohibitive 640,000 solves of the MILP. Second, Q-learning requires diverse samples of the state and action spaces to learn well, but both the state and action are combinatorial.

**Warm starting with off-policy samples**    We design an initialization process, run before the main DQN algorithm (line 2), to help alleviate both of these computational bottlenecks. The main idea is to generate cheap but informative samples which provide rewards observations for a wide diversity of actions, allowing us to warm-start the Q-network much more effectively before training with on-policy samples. We use three strategies for this process.

First, we initialize the Q-network to approximate the single-step (myopic) expected rewards. This leverages our access to a cost-effective simulator for the state transitions and immediate rewards: we can draw a large training set of sampled state-reward pairs from the simulator and fit the Q-network to the immediate reward. Such a simulator is cheap and available, as we are focused here on planning over a known MDP. Training with myopic samples provides an informative initialization when we start learning the long-run rewards.

Second, we draw sampled actions according to the implied myopic policy to seed the training process with a set of "reasonable" actions. Specifically, we use the MILP embedding of the Q-network to find the action maximizing the learned approximation of the single-step reward. This allows the method to work out-of-the-box for different settings, without having to explicitly derive a MILP formulation for the optimal myopic action and incur extra expensive computation from repeatedly solving the MILP. We then train the network using temporal difference updates on the sampled actions and rewards to learn their long-term values.

Third, we introduce diversity into the sampled actions with additional random perturbations. One source of diversity is to randomly flip entries of the myopic action (we call this "perturbed myopic"). Another is to randomly sample *infeasible* actions by, e.g., starting with the all-ones or all-zeros vector for $a$ and then randomly flipping a small number of entries (isolating the impact of including or knocking-out individual actions). This leverages the property that, in restless bandits, *we can simulate valid state transitions even for infeasible actions* because the state transitions are defined independently per-arm. Incorporating perturbed and even infeasible actions greatly increases the diversity of potential samples because for some combinatorial constraints it may be difficult to even find a single feasible solution (and otherwise may be difficult to explore parts of the action space). Similar issues arise if the per-step feasible action space is relatively small (e.g., if the budget is small with $B \ll N$). Including examples of infeasible actions provides a more diverse training set for the Q-network, encouraging better generalization even to feasible actions.

**Variance reduction**    Throughout, we incorporate two additional strategies to lower variance and improve computational efficiency. First, we directly calculate the expected one-step reward (instead of using the observed reward after state transition), which reduces variance (line 10). Second, we memoize the solutions to MILPs seen multiple times in between updates to the Q-network. As we use experience replay with a reasonably small buffer size (10,000) we expect to see repeated samples, enabling us to store the optimal solution to avoid repeated solves. To avoid stale solves, we discard old samples at the same rate that we update the target network $\hat{Q}$.

## 5    EXPERIMENTS

We evaluate the performance of SEQUOIA on the four CORMAB settings introduced in Section 3. In the appendix, we provide further details about the domains (Appendix C), baselines (Appendix D), implementation (Appendix F), and runtime (Appendix G).

**Baselines**    To understand the challenge of combinatorial action selection, we perform an ablation of our method and test ITERATIVE DQN, which leverages the Q-network training used by SEQUOIA to maximize future return. We do not give this method access to a MILP solver, so we instead use a greedy heuristic to overcome the combinatorial action selection problem, greedily selecting the next-best action component (that maximizes the Q-value) while there are still feasible actions. Note that this baseline would not be possible before this paper, because on-policy training of the DQN requires the techniques we developed.

To understand the importance of sequential planning in these settings, we consider three myopic baselines. These myopic solution approaches focus on maximizing the expected reward from the immediate next state, based on the current state and transition probabilities (with no Q-network or any type of rollout). We implement a MYOPIC policy as a MILP that directly encodes (as a linear

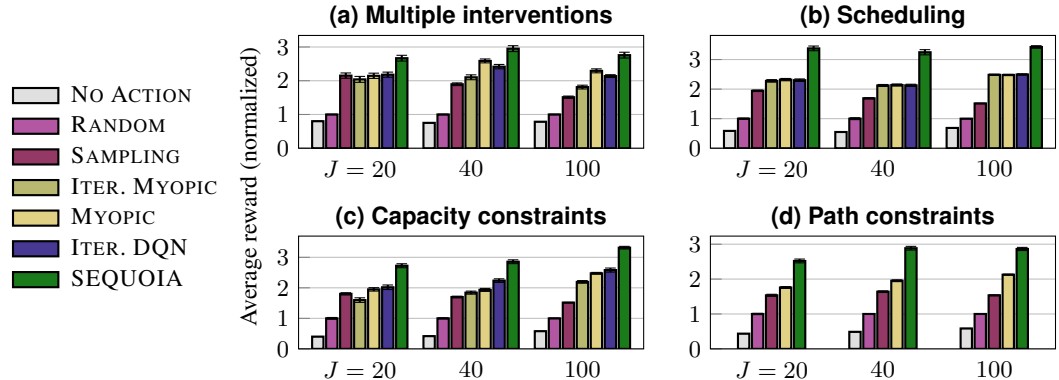

Figure 3: Across all problem settings, SEQUOIA achieves consistently better performance compared to existing methods, which do not consider both combinatorial selection and sequential planning. We evaluate with $J = \{20, 40, 100\}$ arms and $N = \{5, 10, 20\}$ workers. The $y$-axis depicts the average per-timestep reward, normalized to the reward achieved by the RANDOM baseline such that $R_{\text{RANDOM}} = 1$. For the path-constrained problem, note that there are no ITERATIVE MYOPIC or ITERATIVE DQN baselines, as there is no simple iterative approach for selecting a valid cycle.

objective) the expected reward of the immediate next state, while using the same MILP solver as SEQUOIA. MYOPIC would be optimal in a static (non-sequential) setting, but is not necessarily so in the sequential contexts we consider here. We also use a SAMPLING approach, which mirrors the method proposed by He et al. (2016), that randomly samples $k = 100$ actions of the up to $\binom{N}{B}$ possible combinatorial actions and picks the action with the largest expected myopic reward. The final myopic baseline is an ITERATIVE MYOPIC solution construction algorithm that greedily selects one feasible component to build up the full action, similar to ITERATIVE DQN, but using the best single-step reward instead of the predicted Q-value.

Lower bounds on performance are benchmarked by RANDOM which randomly samples a feasible action, and NO ACTION which always takes the null action $a = 0$. We are not aware of additional RL algorithms from the literature that could be applied to CORMAB; this is due to the large combinatorial action space. For instance, many algorithms in the RLlib library (Liang et al., 2018) support discrete actions, but not combinatorial ones.

**Experiment setup**   For each problem instance, we randomly generate transition probabilities, constraints, and initial states. We ensure consistency by enforcing, across all algorithms, that transition dynamics and initial states across every problem instance (across every episode) are consistent across all methods. We evaluate the expected reward of each algorithm across 50 episodes of length $H = 20$ and average results over 30 random seeds. We use Gurobi to solve the MILP, in which we embed a two-layer neural network with ReLU activations. Runtime analysis and implementation details, including hyperparameters, are in Appendix E.

**Results**   SEQUOIA achieves consistently strong performance over the MYOPIC baseline, demonstrating the importance of accounting for long-run return, even in relatively simple MDP settings such as restless bandits. Additionally, note that the MYOPIC approach can only be implemented in settings with simple (e.g., linear or quadratic) constraints and objectives that can be handled by an integer programming solver, whereas SEQUOIA can learn arbitrary objective functions.

To better understand the challenging combinatorial structure of the problem, we look at the iterative baselines which achieve significantly lower performance. Even in the capacity-constrained setting where it performs best, the ITERATIVE DQN approach performs on average 12.1% lower than our non-iterative SEQUOIA—an important takeaway, given that many heuristic approaches for overcoming combinatorial action structure rely on iterative construction heuristics (Dai et al., 2017; Barrett et al., 2020). The SAMPLING heuristic comes close to MYOPIC with 20 arms, but dramatically falls behind at 100 arms—unsurprising, as the action space jumps many orders of mag-

nitude from roughly $\binom{20}{5} = 15{,}504$ actions to $\binom{100}{20} \approx 5.4 \times 10^{20}$ (without considering feasibility constraints). This result underscores the challenge of effectively exploring combinatorially large action spaces.

To show generality and robustness of SEQUOIA, we used the same network architecture and training procedure for all of the different problem settings. Our method works well even without per-domain hyperparameter tuning, which would improve performance further, especially in scaling to larger and more complex problem structures.

## 6 RELATED WORK

**Restless multi-armed bandits**   RMABs generalize multi-armed bandits by introducing arm states that transition depending on whether the arm is acted on. Even when transition probabilities are fully known, computing an optimal RMAB policy is PSPACE-hard (Papadimitriou & Tsitsiklis, 1994) due to the combinatorial state and action space. Heuristic approaches to solve RMABs center around the Whittle index policy (Whittle, 1988; Weber & Weiss, 1990) which uses a Lagrangian relaxation to exploit the fact that the arms are *weakly coupled* (Adelman & Mersereau, 2008; Hawkins, 2003). These relaxations break down in strongly coupled action settings (Ou et al., 2022), our setting here.

Many other solution approaches for RMAB problems are variants of the Whittle index policy, including deep learning to estimate the Whittle index by training a separate network for each arm (Nakhleh et al., 2021); tabular Q-learning (Avrachenkov & Borkar, 2022); and explicitly encoding the Bellman update as a MIP to overcome uncertainty in transition probabilities (Wang et al., 2023). One recent work considers rewards that are not separable across arms, using Monte Carlo tree search to explore arm combinations (Raman et al., 2024), but the action space is still determined by a simple budget constraint. Here, we introduce the first solution approach for restless bandits integrating both deep learning and mathematical programming.

**RL for combinatorial optimization**   An iterative heuristic to solve a static combinatorial optimization problem can be represented as a Markov decision process. RL can then be used to obtain a good policy for constructing a feasible solution. By "static", we mean deterministic problems that are fully specified. This iterative approach have been used by Dai et al. (2017), Barrett et al. (2020), and Grinsztajn et al. (2023); see Mazyavkina et al. (2021) for a more complete survey and Berto et al. (2024) for benchmark problems. The action space in such approaches is *not* combinatorial: to construct a solution to a traveling salesperson problem, one needs to select the single next node to visit in every timestep of the decision process. Breaking with this approach, Delarue et al. (2020) consider combinatorial actions in a capacitated vehicle routing problem, where at every timestep of the decision process a *subset of nodes* that form a tour and respect the vehicle's capacity constraint must be selected. This combinatorial action selection problem is formulated as a MIP whose objective function is the Q-value as estimated by a ReLU neural network. However, their setting is simpler, as they consider a single-shot decision that is deterministic, rather than traditional RL problems which are inherently both stochastic and sequential, such as those that can be represented as an MDP. Additionally, Tang et al. (2020) leverage deep RL for general integer programming solvers to adaptively select cuts, where the action space is the set of all possible Gomory's cutting planes; their approach for addressing the large action space is to sample from the set of feasible cuts according to a specified weighting.

**RL with combinatorial actions**   In the RL literature, it is fairly common to deal with combinatorial *state spaces*, but combinatorial action spaces have received far less attention. A key early example is AlphaGo (Silver et al., 2016), which used deep RL to estimate the probability of winning with each move, combined with Monte Carlo tree search (MCTS) to explore the combinatorial rollout. Feng et al. (2020) apply MCTS and deep RL to deterministic planning, solving 2D puzzles in a deterministic setting.

Dulac-Arnold et al. (2015) deal with large discrete (but not combinatorial) action spaces. Their action selection strategy is sublinear in the number of actions, but this is still prohibitive when the number of actions is exponential as in our setting. He et al. (2016) consider simply sampling a fixed number of actions from the $\binom{N}{B}$ combinatorial action space. Tkachuk et al. (2023) provide a theoretical analysis of RL with combinatorial actions when the value function approximation is

linear in the state–action pair. Because the reward functions in many practical applications may be nonlinear, we opt for neural network function approximations, which are beyond the scope of the aforementioned theory. Brantley et al. (2020) propose an algorithm for budget-constrained continuous action spaces in a tabular RL setting. However, our state spaces are also combinatorial (in addition to our discrete action space), making a tabular state representation impossible. Song et al. (2019) propose to split up combinatorial action selection into iterative non-combinatorial selections; an iterative policy must be learned for this purpose. In contrast, and similar to Tkachuk et al. (2023), we access an optimization solver to find feasible combinatorial actions at each timestep, removing the need for any iterative decomposition of the combinatorial selection as the latter requires learning an additional policy.

**Embedding neural networks in optimization models**   There has been considerable interest in embedding trained neural networks into larger mathematical optimization models; see Huchette et al. (2023) for a survey. Two primary use cases are (1) adversarial machine learning problems, such as finding perturbations of an input that worsen the prediction maximally (Fischetti & Jo, 2018) or verifying network robustness (Tjeng et al., 2018); (2) replacing a function that is difficult to describe analytically with a neural network approximation and then optimizing some decision variables over the approximation (Lombardi & Milano, 2018). As feed-forward neural networks with ReLU activations are piecewise-linear functions in their inputs, they can be represented using a polynomial-size MILP (Fischetti & Jo, 2018; Anderson et al., 2020; Ceccon et al., 2022).

## 7   CONCLUSION

Much recent work has focused on RL *for* combinatorial optimization, but here we address RL *with* combinatorial optimization in the action space. Specifically, we consider offline planning for restless multi-armed bandits where combinatorial action constraints prevent decoupling the problem.

We design our experimental settings around restless bandits because they are an increasingly common planning formulation that requires optimizing for long-term reward in a sequential setting over a discrete, combinatorial action space. However, our method is a general approach for stochastic planning with per-timestep combinatorial actions, provided that the feasible set of actions can be described as an MDP. In particular, our use of model-free RL (via DQN) enables direct adaptation to a wide range of planning problems. Extending SEQUOIA to broader RL planning problems beyond restless bandits would benefit from more efficient methods to explore the large set of state–action transitions, as well as more structured learning methods, such as graph neural networks or other specialized architectures to capture characteristics of the problem setting.

We explored connections to the literature of (stochastic) AI planning, as exemplified by the modeling framework RDDL (Sanner, 2011; Taitler et al., 2023). RDDL is a powerful language for expressing sequential planning problems with known transition dynamics. Of the two direct (i.e., non-RL) solution methods there, the JAX-based gradient ascent method (Gimelfarb et al., 2024) struggles to deal with complex constraints and discrete variables, and the Gurobi-based planner cannot handle stochasticity. However, furthering the connection between the RL-based SEQUOIA and the AI planning perspective is likely to be fruitful.

Another line of exploration deals with further reducing the reliance on potentially expensive calls to a MILP solver. Solving the MILP, both in training the Q-network (Algorithm 1, line 13) and at inference time, is the major computational bottleneck. We use Gurobi to solve the MILP, which uses branch-and-bound to progressively add linear constraints to a continuous relaxation of the problem, but every call to a MILP solver is relatively expensive. We therefore considered solving a continuous nonlinear relaxation of the action selection problem for a faster heuristic approach. We solve these relaxed problems using IPOPT (Wächter & Biegler, 2006), a state-of-the-art nonlinear optimization algorithm that produces stationary continuous action vectors. After a solution has been found, we sample discrete, binary solutions by flipping biased coins parameterized by the IPOPT solution values. On the capacity-constrained CORMAB, this heuristic approach finds significantly better actions than Gurobi, if we allot the same amount of time to each solver. Real-time applications in which decisions must be made instantaneously or in which the number of actions is very large can benefit from this and similar ideas.

ACKNOWLEDGMENTS

Xu acknowledges funding from a Google PhD fellowship and Siebel Scholars. Wilder acknowledges the AI Research Institutes Program funded by the National Science Foundation under AI Institute for Societal Decision Making (AI-SDM), Award No. 2229881. Khalil acknowledges support from the NSERC Discovery Grant program and the SCALE AI Research Chair program.

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

# A  NOTATION TABLE

List of symbols used in the paper and their description.

Table 1: Symbols used in the paper.

| Symbol | Description |
| --- | --- |
| *Problem description* | |
| $s_j \in \mathcal{S}$ | state of arm $j \in [J]$ |
| $a_j \in \mathcal{A}$ | action taken on arm $j$ |
| $P_j(s_j, a_j, s'_j)$ | probability that arm $j$ in state $s_j$ transitions to state $s'_j$ under action $a_j$ |
| $r_j(s_j)$ | reward at time $t$ for arm $j$ in state $s_j$ |
| $R^{(t)}(\boldsymbol{s})$ | reward for restless bandit instance with joint state $\boldsymbol{s}$ |
| $B$ | budget |
| $H$ | time horizon |
| $\boldsymbol{s} \in \mathcal{S}^{\times}$ | joint state of all arms |
| $\boldsymbol{a} \in \mathcal{A}^{\times}$ | joint action of all arms |
| $\mathcal{C} \subseteq \mathcal{A}^{\times}$ | feasible action space, as determined by set of constraints |
| $P^{\times}$ | joint transition probability across all arms |
| $Q(\boldsymbol{s}, \boldsymbol{a})$ | Q-value for action $\boldsymbol{a}$ under state $\boldsymbol{s}$ |
| $V(\boldsymbol{s})$ | value function for state $\boldsymbol{s}$ |
| $\gamma$ | discount factor |
| $E$ | number of episodes |
| $M$ | minibatch size |
| *Notation for* SEQUOIA *algorithm* | |
| $\epsilon$ | epsilon-greedy parameter |
| $\mathcal{D}$ | dataset of historical trajectories stored in replay memory |
| $Q$ | Q-network |
| $\theta$ | Q-network neural network parameters |
| $\hat{Q}$ | target Q-network |
| $\theta^-$ | target Q-network neural network parameters |
| $F$ | target Q-network update frequency |
| *Problem-specific symbols* | |
| $x_*$ | decision variables used in MIP (where $*$ denotes indices) |
| $N$ | action dimension (e.g., number of workers for capacity-constrained) |
| $\phi$ | [*multi-intervention*] link function |
| $\omega$ | [*multi-intervention*] weights of edges for probability transition function |
| $c_i$ | [*multi-intervention & capacity*] cost of action $i$ |
| $b_i$ | [*capacity*] budget constraint for worker $i$ |
| $K$ | [*scheduling*] number of timeslots in schedule |
| $W_{ik}$ | [*scheduling*] matrix denoting whether worker $i$ is available at slot $k$ |
| $A_{jk}$ | [*scheduling*] matrix denoting whether arm $j$ is available at slot $k$ |
| $\mathcal{V}$ | [*routing*] set of vertices in graph |
| $\mathcal{E}$ | [*routing*] set of edges in graph |
| ⌂ | [*routing*] source node for network flow problem |
| $f_{j,k,t}$ | [*routing*] network flow from node $j$ to $k$ at step $t$ |
| $T$ | [*routing*] max path length |

## B    SIMULATION DETAILS

### B.1    THE NEED FOR SEQUENTIAL PLANNING IN RMABS

Myopic policies can perform arbitrarily badly in restless bandits. In Figure 4, we offer a clear example through a restless bandit with two arms, three states, and budget $B = 1$. Suppose the probability $p$ of transitioning right one state is $p = 1$ when the arm is acted upon and $p = 0$ otherwise (in which case the arm will move leftward). If the arms begin at state $s^{(0)} = (0, 0)$, the myopic policy would always act on arm 1, whereas the optimal policy would be to repeatedly act on arm 2. Thus in these experiments, we consider multi-state settings (specifically with $|\mathcal{S}| = 4$ states) to emphasize the sequential aspect of the problem.

Many standard restless bandit problems consider a binary state, binary action setting. The multi-state ($|\mathcal{S}| > 2$) setting enables more realistic modeling of many problems, such as patient engagement or health status.

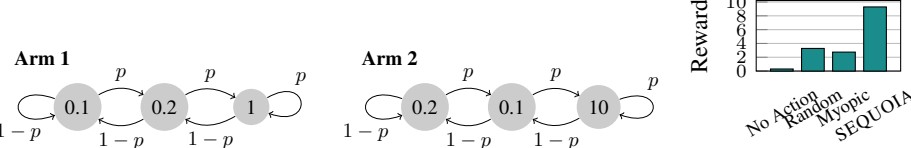

Figure 4: Even in a simple two-arm problem setting with budget $B = 1$, a myopic policy can lead to arbitrarily poor performance in restless bandits. Each arm has three states, with positive reward in each state. Suppose that the probability $p$ of transitioning right one state is $p = 1$ when the arm is acted on and $p = 0$ otherwise. This problem instance leads to the rewards on the right, where the gap between myopic and SEQUOIA can be arbitrarily large depending on the rewards at the rightmost state. This poor performance results even in this simple problem setting.

### B.2    BASE MODEL: RESTLESS BANDIT WITH 4 STATES

To evaluate our sequential planning algorithm, we design a simulator that is designed to ensure the myopic policy is non-optimal. Thus across all experimental settings, we consider restless bandit problems with $|S| = 4$ states, where a fraction of the arms are long-run beneficial to act on (but myopically non-optimal), and the remainder of the arms are short-run beneficial to act on but myopically non-optimal. This base restless bandit problem is implemented as the `MultiStateRMAB` class.

In these four states $s \in \{0, 1, 2, 3\}$, we model that users can only transition between consecutive states in a single timestep; for example, they cannot skip from state $s = 1$ to state $s' = 3$. We denote moving up a state as $s^+$ and moving down as $s^-$. For completeness: $s^- = \max\{0, s - 1\}$ and $s^+ = \min\{3, s + 1\}$.

The bad arms have the following reward for each of the four states: $[0.1, 0.15, 0.2, r_{\text{bad}}]$ where $r_{\text{bad}}$ is drawn randomly from $\{4, 5, 6\}$, and the good arms have reward $[0.2, 0.15, 0.1, r_{\text{good}}]$ where $r_{\text{good}}$ is drawn randomly from $\{1, 2, 3\}$.

For the bad arms, the transition probability of moving up a state is drawn uniformly at random between $[0.0, 0.2]$, and the probability of moving down a state is drawn from $[0.7, 0.9]$.

## C    DOMAIN DETAILS

### C.1    PATH-CONSTRAINED

We encode the constraints of this path-planning problem as flow constraints on a time-unrolled graph. We allow the maximum path length $T$ to be double the budget, that is, $T = 2 \cdot B$. All arms pulled must fall along the path visited; we can pull at most $B$ arms out of at most $T$ visited.

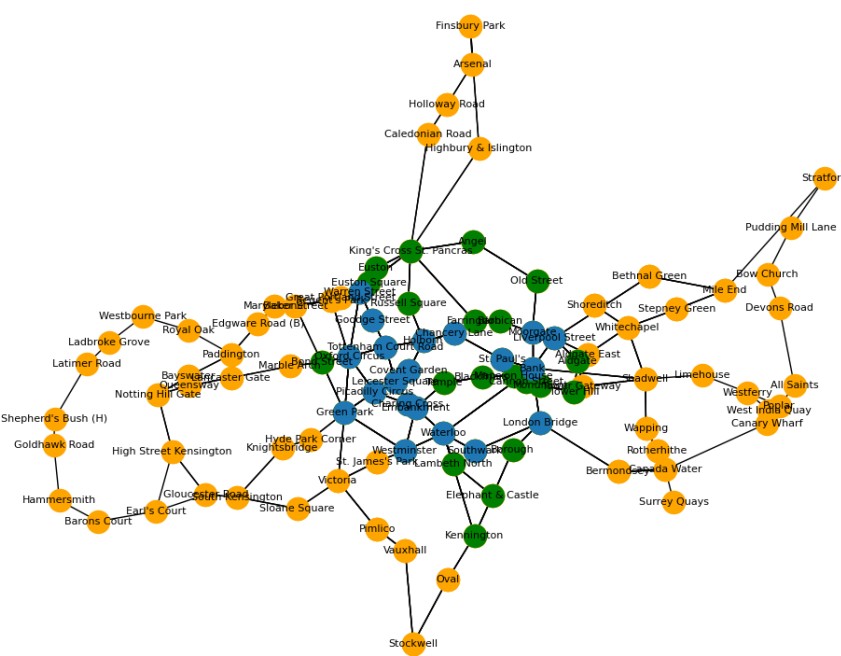

Figure 5: Graph used for the path-constrained problem, based on a diagram of the London underground. Blue represents nodes used for $J = 20$. Green nodes are added for $J = 40$, and orange nodes are added for $J = 100$.

We introduce flow variables $f_{j,k,t}$, which equal 1 if we take the edge from node $j$ to $k$ as the $t$-th segment of our path and 0 otherwise. Thus for every edge $(j, k)$ in $\mathcal{E}$, we have decision variables $f_{j,k,t}$ and $f_{k,j,t}$ for all $t \in \{1, \ldots, T\}$. Node $\text{⌂} \in \mathcal{V}$ is the source, where we must begin and end the path. Finally, decision variable $a_j$ indicates whether we act on arm $j$, i.e., we pass through node $j$.

$$\max_{\mathbf{f}, \mathbf{a}} \ Q(\mathbf{s}, \mathbf{a}) \tag{7}$$

$$\text{s. t.} \sum_{k:(\text{⌂},k) \in \mathcal{E}} f_{\text{⌂},k,1} = 1$$

$$\sum_{k:(k,\text{⌂}) \in \mathcal{E}} f_{k,\text{⌂},T} = 1$$

$$\sum_{(j,k) \in \mathcal{E}} f_{j,k,t} = 1 \qquad\qquad \forall t \in [T]$$

$$\sum_{k:(k,j) \in \mathcal{E}} f_{k,j,t} = \sum_{k:(j,k) \in \mathcal{E}} f_{j,k,t+1} \qquad\qquad \forall j \in \mathcal{V}, t \in [T-1]$$

$$a_j \le \sum_{t \in [T]} \sum_{k:(j,k) \in \mathcal{E}} f_{j,k,t} \qquad\qquad \forall j \in \mathcal{V}$$

$$\sum_{j \in \mathcal{V}} a_j \le B$$

$$f_{j,k,t} \in \{0,1\} \qquad\qquad \forall j, k \in \mathcal{V}, t \in [T]$$

$$a_j \in \{0,1\} \qquad\qquad \forall j \in \mathcal{V}$$

**Implementation** The underlying graph is based on a real-world network of the London tube, where one node is placed at each station.[1] We have that the graph is fully connected. We visualize

---

[1]Map data is available on the Github repo, and comes from `https://commons.wikimedia.org/wiki/London_Underground_geographic_maps`

this graph in Figure 5, where we color the subset of nodes used for the three problem settings $J = \{20, 40, 100\}$. For simplicity, we consider the cost of each edge to be 1, but heterogeneous costs on the edges could be easily implemented.

Due to the significantly increased complexity of the MILP for this path-constrained problem, due to having the encode the time-unrolled graph with $NJT$ nodes, we take advantage of the decoupled reward structure for this problem in the training process. That is, in the training loop (lines 9 and 13 in Algorithm 1) we impose only the budget constraint and not the path constraint. This significantly speeds up the training time, as for the setting with $J = 20$, the MILP is simplified from 800 variables and 213 constraints to only 20 variables and 1 constraint. In the setting with $J = 100$, the reduction is even greater, from 14,740 variables and 4,043 constraints to 100 variables and 1 constraint.

Note that for all the problem instances we describe here, with the exception of the multi-action setting, the reward structure is decoupled, as each action impacts the transition probability for only a single arm. Thus this heuristic training process can be used for all settings in which the reward structure can be decoupled.

### C.2 SCHEDULING

We have $N$ workers and $J$ arms (e.g., patients). Each worker has some set of available timeslots, and each patient has their own set of available timeslots, out of $K$ timeslots. We can assign at most one worker to one patient for each timeslot, and each patient must be intervened on by two compatible workers. We can intervene on at most $B$ patients.

This scheduling problem can be formulated as follows:

$$
\max_{\mathbf{x}, \boldsymbol{a}} \ Q(\boldsymbol{s}, \boldsymbol{a}) \tag{8}
$$

$$
\text{s.t.} \ \sum_{j \in [J]} a_j \leq B
$$

$$
\sum_{j \in [J]} x_{ijk} \leq 1 \qquad\qquad \forall i \in [N], k \in [K]
$$

$$
\sum_{i \in [N], k \in [K]} x_{ijk} \leq 2 \qquad\qquad \forall j \in [J]
$$

$$
2 \cdot a_j \leq \sum_{i \in [N], k \in [K]} x_{ijk} \qquad\qquad \forall j \in [J]
$$

$$
a_j \geq x_{ijk} \qquad\qquad \forall i \in [N], j \in [J], k \in [K]
$$

$$
a_j \in \{0, 1\} \qquad\qquad \forall j \in [J]
$$

$$
x_{ijk} \in \{0, 1\} \qquad\qquad \forall i \in [N], j \in [J], k \in [K]
$$

The scheduling constraints are encoded as binary matrices, where $W_{ik}$ and $A_{jk}$ denote whether resource $i$ (or patient $j$) is available at timeslot $k$. We introduce a decision variable $x_{ijk}$ whenever a worker $i$ and patient $j$ are both available at timeslot $k$, so $x_{ijk}$ exists iff $W_{ik} = A_{jk} = 1$. Then, $x_{ijk} = 1$ indicates the assignment of worker $i$ to arm $j$ at timeslot $k$, and $a_j$ indicates whether we "pull" arm $j$. Each worker can only be assigned once per timeslot.

This problem also generalizes the standard RMAB: each pulling action requires solving a matching problem to assign workers to each action. The standard RMAB problem would correspond to the setting where all workers and all patients are available for all timeslots (for simplicity, there can be just a single timeslot during which all workers and all patients are available), and each patient must be intervened on by just one worker. In this case, the number of workers simply represents the budget constraint.

**Implementation**  We use $K = 5$ timeslots in all problem settings. We randomly select 2 timeslots for each arm and 3 timeslots for each worker.

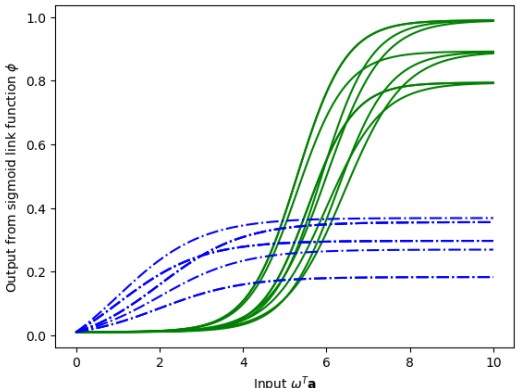

Figure 6: A visualization of sigmoid functions used for the Multiple Interventions setting, where values of $a$, $b$, and $x_0$ are randomly generated. We consider actions with $\omega = 2$. In this setting, for the arms with reward shown in dashed blue, $\omega = 2$ would have stronger effect for the first two actions deployed, but their impact would be relatively limited. In contrast, for arms with transition probability shown in solid green, the first two actions applied to an arm would lead to relatively low transition probability, but the third actions would lead to a significant increase, and in all cases an increase greater than three actions applied to any of the blue arms.

## C.3 CAPACITY-CONSTRAINED

We have a decision variable $x_{ij}$ for all $i$ and $j$, indicating whether worker $i$ gets assigned to arm $j$.

$$\max_{\mathbf{x}, \boldsymbol{a}} \ Q(\boldsymbol{s}, \boldsymbol{a}) \tag{9}$$

$$
\begin{aligned}
\text{s.t.} \ & \sum_{j \in [J]} c_j x_{ij} \leq b_i && \forall i \in [N] \\
& a_j \leq \sum_{i \in [N]} x_{ij} && \forall j \in [J] \\
& x_{ij} \in \{0, 1\} && \forall i \in [N], j \in [J] \\
& a_j \geq x_{ij} && \forall i \in [N], j \in [J] \\
& a_j \in \{0, 1\} && \forall j \in [J]
\end{aligned}
$$

The standard RMAB setting would correspond to a single worker ($N = 1$) with a budget $b_i = B$.

**Implementation**  We generate constraints by selecting random costs for each arm ($c_j \in \{2, \ldots, 6\}$ and random capacity for each worker ($c_j \in \{2, 7\}$.

## C.4 MULTIPLE INTERVENTIONS

In the public health literature, these link functions have often been described as S-shaped, such as in a widely used model of smoking cessation (Levy et al., 2006) The S-shape implies that as individuals are impacted by more actions, they first experience increasing returns in their probability of transition to an improved state (the first part of the S) before plateauing as the effect saturates (the second part). While realistic, S-shaped curves are often NP-hard to approximately optimize because they violate the diminishing returns assumptions required for submodular optimization (Schoenebeck & Tao, 2019). As we are motivated by public health interventions, we use these as motivating descriptions in the descriptions, but these problem structures exist in many other applications. Importantly for public health and other applications, these constraints could be further specified incorporate additional desiderata, such as fairness constraints. For example, if we had demographic features associated with each patient, we could encode a requirement to visit at minimum some fraction of each subgroup, such as the most elderly.

The sigmoid curve we use is of the form:

$$\phi(x) = a \cdot \frac{1}{1 + \exp(-b \cdot x - x_0)} \tag{10}$$

where values of $a \in (0, 1]$, $b \in [1.4, 2]$, and $x_0 \in \{0, \ldots, 10\}$ are randomly generated for each arm. Instantiations of the sigmoid curves we use in the simulation is visualized in Figure 6.

Note that the implemented class here is designed to also adapt to a linear link function, which could be a useful model for some problems. The setting with a linear link function would be simpler to solve; it would not require solving for combinatorial actions as an iterative approach would be sufficient.

**Implementation** We consider $N = 2 \cdot B$ actions, of which only $B$ can be pulled in each timestep. For the purposes of evaluation in this setting, we considered a relatively simple setting where each action is connected to up to 3 arms. More intricate configurations could benefit from more sophisticated learning approaches that incorporate the structure of the problem setting, such as graph neural networks (GNNs).

## D    BASELINES

### D.1    NO ACTION BASELINE

At every timestep, the action taken is the null action ($a_j = 0$) on every arm.

Since we impose the assumption that actions only have positive impact on reward, this baseline serves as a lower bound on the possible reward in each setting.

### D.2    RANDOM BASELINE

At every timestep, we take one random valid action. Thus the action we take at every step is independent of the current state of the RMAB.

The random actions selected are:

- **Multiple interventions**: Select a random subset $\binom{N}{B}$ of the actions, according to our budget.

- **Capacity constrained**: Randomize the order of workers and arms (patients). For each worker, go through the list of arms and greedily select each one, while that worker has remaining capacity. Continue until either the worker's capacity is exhausted or we have iterated through all arms.

- **Scheduling**: Randomize the order of workers and patients. Iterate through patient, and find the first two compatible workers that have not yet been assigned, if any exist. If such workers exists, assign those workers to the patient, then continue on to the next patient.

- **Routing**: Generate all the simple random paths within the path constraint $T = 2 \cdot B$, using the `simple_cycles` function of the `networkx` package. Pick one of these paths at random. Then select a random subset $\binom{T}{B}$ of the nodes within that path to act on.

  Note that, because we are limiting ourselves to all *simple* paths (no repeating vertices), this approach is weaker than picking at random from among all the valid actions. A stronger approach would require significantly more complexity, as finding a loop of at most length $T$ is by itself an NP-hard problem.

### D.3    SAMPLING BASELINE

Sample $K$ random actions ($K = 100$ by default), evaluate a few rollouts with that action, and select the action that induces the great observed reward.

### D.4 MYOPIC BASELINE

Solves the MILP using expected value (based on the true transition probabilities) from the immediate next timestep, without doing any long-term planning. Otherwise, we precisely solve the MILP to get an optimal myopic action.

- **Multiple interventions**: For the MILP myopic solver, we encode the sigmoid curve with a piecewise-linear approximation, which can be represented as a constraint. In the general case, PWLs can approximate continuous functions well[2], and can be directly resolved in Gurobi[3].
- The other settings use the same MILP as described above, using the expected value of the next immediate timestep using true transition probabilities.

### D.5 ITERATIVE MYOPIC BASELINE

- **Multiple interventions**: Evaluate each action, and pick the action that brings the biggest increase in reward.
- **Capacity constrained**: Go through the workers in ascending order of their capacity. For each worker, pick from among the arms that we can afford the arm that yields the best ratio in terms of relative increase in value. Continue until the workers' capacity is lower than the cost of any available arm.
- **Scheduling**: Organize workers by descending order of the number of available slots. For each worker, evaluate the value of adding each arm, if the worker and that arm have a compatible timeslot.
- **Routing**: No computationally efficient solution that is iterative; not implemented.

### D.6 ITERATIVE DQN

Same as the iterative myopic baseline, but instead of using the expected value of the one-step next state, we use the Q-network estimates to estimate the long-run value.

## E EXPERIMENTAL DETAILS

### E.1 DATA GENERATION

Across all experimental settings, the base MDP used to represent each arm is the 4-state MDP described in Appendix B.2. We randomly generate transition probabilities. The constraints on the actions are generated for each problem setting as described in Appendix C.

## F IMPLEMENTATION

Our implementation uses PyTorch and TorchRL (Bou et al., 2024) libraries. We build off code from Dumouchelle et al. (2022) to embed a neural network with single-layer ReLU activations as a MILP (Fischetti & Jo, 2018).

For consistency in results, the random seeds are set between 1 and 30 for the results.

To train the Q-network, we use the hyperparameters specified in Table 2.

## G RUNTIME

Experiments are run on a cluster running CentOS with Intel(R) Xeon(R) CPU E5-2683 v4 @ 2.1 GHz with 8GB of RAM using Python 3.9.12. The MIP was solved using Gurobi optimizer 10.0.2.

---

[2]Francis J. Narcowich notes `https://www.math.tamu.edu/~fnarc/m641/m641_notes/continuous2014.pdf`

[3]`https://www.gurobi.com/documentation/current/refman/cs_model_agc_pwl.html`

Table 2: Hyperparameters used in implementation to train SEQUOIA

| Hyperparameter | Setting |
|---|---|
| Discount factor $\gamma$ | 0.99 |
| Epsilon greedy $\epsilon$ | 0.9 starting; final 0.05; 1000 rate of exponential decay |
| Learning rate | $2.5 \times 10^{-4}$ |
| Adam $\epsilon$ | $1.5 \times 10^{-4}$ |
| Target update rate $\tau$ | $1 \times 10^{-6}$ |
| Minibatch size | 32 |
| Replay memory size | 10,000 |
| Number of training episodes | 100 |
| Network architecture | Two hidden layers of size 32 each |

Empirically, we noticed that a major computational bottleneck was the size of the Q-network. Even with only two hidden layers, training a network with 128 hidden units took an order of magnitude longer than with 32 hidden units, underlining the tradeoff between expressivity of the Q-network and runtime.

The average running time of the training procedure of SEQUOIA is listed in Table 3.

Table 3: Total runtime (in minutes)

| $J = 20$ | $J = 40$ | $J = 100$ |
|---|---|---|
| 25.5 | 76.7 | 475.2 |

