# OpenReview forum: "Reinforcement learning with combinatorial actions for coupled restless bandits"
_ICLR.cc/2025/Conference — ICLR 2025 Poster_

### Official Review · Reviewer_9UJj · 2024-10-29

**Soundness:** 3
**Presentation:** 3
**Contribution:** 2
**Rating:** 6
**Confidence:** 4

**Summary:**

This paper propose a more general restless bandit model---coRMAB, in which the action space for different arms could also be correlated (e.g., one action can influence multiple arms). In this model, the authors adapt the idea of DQN, and utilize the fact that solving integer programming with a feed-forward neural network programming representation is efficient. They propose the SEQUOIA algorithm, and show that it achieves good performances in experiments.

**Strengths:**

- The problem setting is well-motivated.

- I quite like the idea that instead of solving the exact combinatorial optimization, we choose to solve the optimization based on our estimation as an approximate approach.

- From experiments, this idea seems work well.

**Weaknesses:**

- There are no real data experiments. For a paper that does not contain too much theories, I believe real data experiments are necessary.

- There are some parts that are not very clear to me, e.g.,

In Eq. (2), why there is no $s'$ in RHS?

For "Schedule-constrained", "Capacity-constrained", and "Path-constrained", are there any formulation about the transitions?

Why we only consider these four kinds of cdRMAB? I think your algorithm (or solving the MILP) is not restricted to these four settings, right?

In line 431-432, it is said that "For example the ITERATIVE myopic approach performs on average 14.6% lower than optimal MYOPIC". But I do not see that? In Figure 3(b) and 3(c), they are very close, and in Figure 3(a), it seems that ITER.MYOPIC is higher than MYOPIC?

**Questions:**

See "Weaknesses" for details.


======After rebuttal=======

Thanks for the reply. I do not have further questions.

---

> ### Author Response · Authors · 2024-11-13
>
> Thank you for your positive review! We are glad that you see our paper is well-motivated, and that our approach of solving the exact optimization problem is better than an approximate approach.
>
> ## Response to Weaknesses/Questions
>
> 1. **Real data experiments:** We agree that it would be preferable to use real data. However, we do not know of publicly available data sources for a restless bandit setting. We hope this changes in the near future!
>
> 2. **Eq. (2):** There is no $s’$ in the RHS because the equation represents the probability of transitioning to a higher-reward state — we have clarified this in the paper.
>
> 3. **Transition probability formulation:** Yes, the transition dynamics for these settings are specified in Appendix B.2.
>
> 4. **Why 4 settings:** We introduced 4 new formulations of coRMAB that we think are practical representations of many real-world settings (with multiple types actions and with path, capacity, schedule constraints).
>
> Of course, our approach is generalizable to any deep RL setting where the actions (either continuous or binary) have constraints that can be formulated as a mixed-integer program. This of course generalizes to a very wide range of problems.
>
> 5. **Iterative approach:** Thank you for catching this typo! This was supposed to read “Iterative DQN” and “SEQUOIA”, not referring to Myopic. We have corrected this.

---

### Official Review · Reviewer_ya5G · 2024-11-04

**Soundness:** 3
**Presentation:** 3
**Contribution:** 2
**Rating:** 6
**Confidence:** 4

**Summary:**

The work introduces a new class of multi armed bandits problem, coRMAB which generalizes Restless bandits problem where the arm action cannot be decoupled because of the constraint of the problem that is common in real world scenarios. The authors also briefly go through the four scenarios with valid examples and propose an algorithm SEQUOIA based on deep RL algorithm – Q learning & mathematical optimization to optimize long-term reward. The authors also highlight the issue with very large action space and showcase the ability of SEQUOIA to perform on those scenarios with experiments comparing them with some of the other algorithms that can handle this problem.

**Strengths:**

1.	The problem setting and formulation are interesting. The formulations discussed in this paper is a natural and general extension of restless bandits. Most of the real-world scenarios often fall under one of the four scenarios highlighted by the authors in this work.

2.	Deep Q learning type algorithms are generally computationally heavy, and it only grows with the action and state space. This work addresses this issue and takes this into account in their problem formulation and algorithm.

3.	A key challenge in using deep learning within a RL problem or any problem in general is the need for optimizing the network architecture and resource for hyper-parameter tunning and the algorithm seems to work with minimal alterations across domains.

**Weaknesses:**

1.	The work highlights the empirical results of the proposed algorithm SEQUOIA but it did not have any theoretical guarantees on measures like Regret or convergence bounds. Having them would have greatly benefitted the solidarity of the developed algorithm.

2.	The major results shown in this work is about the experiments and how SEQUOIA, the developed algorithm performs on four scenarios of the problem formulation and competes with some of the other algorithms that can be modified to work on coRMAB, however a detailed experimental design could be carried out to further showcase the benefits and limitations of the proposed algorithm and how they perform in different regimes on different transition dynamics.

3.	The work also assumes that it is an offline planning setting, i.e., the transition dynamics are known in advanced which can be a limiting factor on many practical settings where the transition dynamics are harder to compute. Most of the real-world setting involves an agent interacting with an environment to understand them.

**Questions:**

1.	The problem of coRMAB extends the problem of restless MAB to handle actions that cannot be decoupled. If we were to set the no of actions (N) equal to the no. of arms (J)  and using a simple budget constraint where \sum j \in [J] a_j <= B and also making each action only connect to its corres. arm, we end up in restless MAB setting. In that case, How does SEQUOIA handle the Restless bandit problem ?

2.	For the case of standard restless bandit problem, how does SEQUOIA competes with some of the existing algorithm in the space of restless bandit problem like restless-UCB [Reference B], which tends to have a sublinear Regret bound with good empirical performance on real-world data too. ?

3.	Also, a detailed comparison of the algorithm with other algorithms or approach could help better understand the performance of the developed algorithm. For instance, a comparison of SEQUOIA with other algorithm/ approach on the basis of either Regret/ Normalized Average reward would better help understand the performance gain of the proposed algorithm. ?

4.	The metrics used in this paper is normalized average reward. Given that we know the transition dynamics, does comparison against the optimal best policy performance be a better metrics like Regret ? Or Other convergence guarantees like one shown in this paper [Reference A] be better fitted to potentially quantify the significance of this work ?


5.	Also, solving the large action space problem is computationally hard, however how does the complexity grows if we were to increase the neural network size for a more complex system ?

6.	Also, the SEQUOIA uses the same network architecture across all the four constraint type proposed in the paper, Is it optimal or does tunning the hyper-parameter for each constraint provide better performance ?

7.	How does SEQUOIA’s result compare to existing domain specific solution for the four-constraint type setting discussed. This would better help SEQUOIA solidify its performance gain with better clarity ?

Reference:
A.	Guojun Xiong, Jian Li, Finite-Time Analysis of Whittle Index based Q-Learning for Restless Multi-Armed Bandits with Neural Network Function Approximation, Advances in Neural Information Processing Systems 36 (NeurIPS 2023)
B.	Siwei Wang, Longbo Huang, John C. S. Lui, Restless-UCB, an efficient and low-complexity algorithm for online restless bandits, Advances in Neural Information Processing Systems (NeurIPS 2020)

---

> ### Author Response · Authors · 2024-11-13
>
> Thank you for your positive review and suggestions! We are glad you find our problem formulation interesting and realistic to the real-world, and our solution practical for addressing the computational intensity of deep RL.
>
> We will update the paper to reflect the following response.
>
> ## Response to Questions
>
> 1. **Handling standard restless bandits:** Indeed, our formulation of coRMAB generalizes the standard RMAB. We mention this in lines 175-178, 944, and 971-973 for our formulation of different settings.
>
> 2. **Comparison to Restless UCB:** Restless UCB focuses on a learning problem with unknown transition dynamics, which is a different setting. We agree they make a valuable contribution for learning in RMABs. However, their experiments only scale to $N=5$ arms and requires over $T=50,000$ timesteps to learn — clearly, that is not practical for real-world problem settings.
>
> 3. **Comparison to other algorithms:** Regret would be a useful metric to compare if we are considering a learning setting with unknown transition dynamics. However, here instead we focus on planning with known transition dynamics, where the challenge is overcoming computational complexity, not unknown dynamics.
>
> 4. **Comparison to the optimal policy:** Please see response to Question #3 for why we cannot consider regret.
>
> 5. **Scalability for larger neural networks:** Yes, the complexity does grow as the neural network for DQN grows; we say in lines 291-292 that the MILP requires a linear $O(DP)$ binary variables and constraints where $D$ is the number of hidden layers and $P$ is the number of neurons per layer.
>
> 6. **Effect of hyperparameter tuning:** Our results are shown without hyperparameter tuning. We expect that the empirical performance could improve further with more tuning.
>
> 7. **Comparison to domain-specific solutions:** There are not existing domain-specific solutions for the sequential, combinatorial-action problem we consider, so we cannot compare to them. We introduce the coRMAB problem in this paper.
>
> ## Response to Weaknesses
>
> 1. **Theoretical guarantees:** Please see response to Question #3 for why we cannot evaluate regret.
>
> For other converge bounds, note that the vast majority of deep RL approaches focus on empirical performance, not on theoretical guarantees. For example, DQN (the deep RL algorithm we use) was introduced in 2015, but the first attempt for a theoretical analysis of DQN was not done until 2020 [5] — and this paper makes extremely simplifying assumptions, such as assuming the data-gathering policy has good coverage (thus requiring minimal exploration) and that the trained neural network is sparse (i.e., mostly 0s).
>
> Our focus is on practical restless combinatorial bandit problems. We have therefore proposed realistic problems and realistic generation schemes.
>
> 2. **Experimental design:** Please see response to Question #7
>
> 3. **Assumption of an offline planning setting:** Restless multi-armed bandits are designed as planning problems; see Whittle [1988] and Weber and Weiss [1990], and restless bandits that assume known dynamics have been applied to several real-world settings; see Raman et al. [2024] for food rescue and Mate et al. [2022] for public health in India.
>
> ### References
> > Mate, et al. Field study in deploying restless multi-armed bandits: Assisting non-profits in improving maternal and child health. AAAI 2022
> >
> > Raman, Shi, and Fang. Global rewards in restless multi-armed bandits. NeurIPS 2024
> >
> > Whittle. Restless bandits: Activity allocation in a changing world. Journal of Applied Probability 1988
> >
> > Weber and Weiss. On an index policy for restless bandits. Journal of Applied Probability 1990

---

> > ### Comment · Reviewer_ya5G · 2024-11-27
> >
> > Thank you for responding to my comments. The authors response to Q2 (comparison with restless UCB) and Q3&Q4, comparison with existing algorithm have clarified my initial concerns. The authors have provided relevant information to clarify some of my initial concerns in their work related to the computational efficiency. I suggest including this information into the next version of their manuscript for a better clarity in this work.

---

### Official Review · Reviewer_EMxA · 2024-11-05

**Soundness:** 3
**Presentation:** 3
**Contribution:** 2
**Rating:** 6
**Confidence:** 3

**Summary:**

This paper addresses a RL problem setting where the action space is combinatorial and discrete, I.e. in which actions are coupled with combinatorial constraints. This work uses the formalism of restless multi-armed bandits to tackle the problem and consider the setting where arms are coupled which leads to a large action space using 4 different examples: multiple interventions, path constraints, bipartite matching and capacity constraints. The proposed approach relies on embedding a Q-network into a mixed integer program fro combinatorial action selection at each time step. The proposed RL algorithm SEQUOIA optimizes for long-term reward over the action space allows to perform sequential planning for a combinatorial action space setting. The performance of the algorithm shows empirical improvement over existing approaches.

**Strengths:**

- The 4 example settings provided are compelling for motivating the work. I find the public healthcare example interesting and it is nicely used as a running example. I believe these sequential planning problems are important problems to solve in practice.
- Writing is clear overall and the presentation with the figures is nice.

**Weaknesses:**

- The mathematical formulation of the problem is a bit confusing and could be more rigorous:
(a) Is it an infinite horizon problem (since you seem to be using discounting)?
(b) l. 99: you mention that your approach enables per-timestep combinatorial action spaces. Where does this flexibility show up in the problem formulation in sec. 2.2. The set C is a fixed combinatorial action vector set. I do not see any time dependence taken into account in the formulation, C also seems to be fixed in Algorithm 1. Usually in RL, action sets are fixed. In your setting, it might be useful to consider time varying ones as the actions that might be available might change because of the coupling of the actions, for instance reducing over times due to the previous actions chosen that limit the remaining possible choices.

- Transition dynamics and rewards are assumed to be known a priori (l. 104). This might be quite limiting regarding the health care motivating example.

- About the comparison to standard DQN (Fig. 2 + l. 243-247): in standard DQN the output size of NN scales with the size of the action space. In your approach, now the input size has to be of the size of the action space (which is exponentially large) to be able to encode any action input from the large combinatorial space you consider. Any comment about this? Why is it more tractable as for the main scaling challenge you want to overcome?

- As discussed in l. 319-323: having to solve an MILP for each sample and for each time step seems extremely expensive.

- Q learning has even been used for continuous actions spaces via appropriate discretization of the action space.  I believe stronger and more convincing arguments have to be made here to support the claims of the paper since this is a crucial point given the motivation of the paper. Could you please elaborate and clarify better what makes your approach scalable compared to prior existing algorithms applied to your combinatorial action space setting? See follow-up question below.

- As discussed in the paper, the idea of embedding a neural network into a mixed-integer problem is not new. Could you elaborate more on the technical challenges faced when following this approach and why does it address the scalability challenge in your problem?

Minor: l. 1028: seems empty for ‘Multiple interventions’, any missing description here?

**Questions:**

**Main questions:**
- How crucial is the assumption of known dynamics and rewards for your approach?
- Running time: Table 3 in the appendix shows the total running time depending on the number of arms. What about the number of workers? Is it fixed in this table?
- Why is step 9 involving an argmax over actions more tractable than given the combinatorial nature of the action space? This is an important point for scalability that is not very clear to me from the presentation.
- What’s the size of the combinatorial action set in the experiments for each of the 4 examples? Why is it prohibitive for existing RL methods?
- l. 345 ‘We introduce diversity into the sampled actions with additional random perturbations’. It seems that there is no way to bypass the need to see a sufficient number of diverse actions. I guess this is also an exploration requirement to solve the RL task. If you cannot explore a large number of actions, I guess there is little that can be said about the quality of the obtained policy.
- Can you further justify the use of DQN? I understand that this is probably the most famous one but since you consider a known transition model, would DDPG make also sense to be tested?
- Why don’t you compare to the approach you mention in l. 494-498?


**Minor questions:**
- Why do you need a piecewise linear approximation of the sigmoid link function (l. 181) which is known and can be computed?
- Any interpretation for including self-loops (l. 196)?

---

> ### Author Response · Authors · 2024-11-13
> **Response pt 1**
>
> Thank you for your review and suggestions! We are happy that you find our work to be compelling and well-presented. We will update the paper to reflect the following response.
>
> ## Response to Main Questions
> 1. **Assumption of known dynamics:** We focus on RL for planning, so we do assume known dynamics and rewards. Restless multi-armed bandits are designed as planning problems; see Whittle [1988] and Weber and Weiss [1990], and restless bandits that assume known dynamics have been applied to several real-world settings; see Raman et al. [2024] for food rescue and Mate et al. [2022] for public health in India.
>
> 2. **Running time Table 3:** In this table we fix the number of workers to $N=10$ (the middle setting in Figure 3).
>
> 3. **Tractability in combinatorial action space (step 9):** Because our action spaces are combinatorial, a DQN-type network with as many outputs as the number of actions is not feasible. In other words, one cannot simply input the state vector into a network and read out Q(s,a) for all possible actions $a$ (top-left in Figure 2). To get around this, we use a network which takes as input a pair $(s,a)$ and estimates $Q(s,a)$ only for that pair. It remains to find the action $a$ which maximizes the output of this network for a fixed state s; this is what MILP solving does.
>
> 4. **Size of the combinatorial action sets, and why is it prohibitive for RL:** For the experiments, with the largest setting ($J=100$ arms and $N=20$ workers) the number of possible actions is $\approx 5.3 \times 10^{20}$; this is clearly prohibitive for existing RL methods.
>
> 5. **Need for diverse actions:** We do need to explore a number of diverse actions, but as we discuss in lines 346–350, we are able to efficiently do so in this setting.
>
> 6. **Why DQN not DDPG:** Given that we have to solve a MIP to evaluate our policy, we chose to use DQN as it is an effective and simple RL approach that requires only a Q-network, whereas DDPG requires both an actor and a critic.
>
> 7. **Comparison to approach in l.494-498:** Tkachuk et al. (2023) assume linear Q-realizability (Assumption 1 in their paper), which requires that the true $Q(s,a)$ can be approximated by a linear function in a feature vector representing the state-action pair $(s,a)$. This is a restrictive assumption which we do not make. Additionally, they assume that the actions are continuous rather than discrete, which they clarify: “​​​​Combined with the linear $Q \pi$-realizability (Assumption 1), the greedy oracle amounts to solving a linear optimization over the action set $A$.’’ This is a much simpler setting than ours that is amenable to theoretical analysis.
>
> ## Response to Minor Questions
>
> 1. **Need for PWL approximation of sigmoid:** A mixed-integer linear program cannot directly model a continuous non-linear sigmoid function. A piecewise-linear approximation is MILP-representable.
>
> 2. **Self-loops in path-constrained coRMAB (l. 196):** The self-loops ensure that all paths of total length $\leq B$ are valid. If the budget is $B=10$, the reward-maximizing path might be of length $9$ — but without self-loops, a path of length $10$ might not be feasible (because we would have to go to another node and come back, requiring 2 extra edges).

---

> ### Author Response · Authors · 2024-11-13
> **Response pt 2**
>
> ## Response to Weaknesses
>
> 1. **Clarifications on problem formulation:** (a) Indeed, this is an infinite-horizon setting; the horizon $H$ is used as the number of timesteps in evaluation. We clarified in the updated submission. (b) By “per-timestep” in l.99, we simply meant that the action space is combinatorial. We agree that SEQUOIA can handle time-varying action spaces by appropriately restricting the available actions per timestep in the MIP model.
>
> 2. **Transition and rewards assumed to be known:** Please see response to Question #1
>
> 3. **Comparison to standard DQN and combinatorial action space:** Precisely as you say, with standard DQN the output size of the NN scales with the size of the action space, which is exponentially large. That is the key contribution of our paper — to directly solve using an NN whose output size is only the dimensionality of the action space.
>
> 4. **Cost of solving MILP in every timestep / scalability:** Selecting an action from a constrained combinatorial set necessitates some form of combinatorial optimization. MILP solving is one generic way of doing so as it can simultaneously represent the trained ReLU Q-network as well as the combinatorial constraints. Optimizing (1) a deep network objective function with (2) binary variables and (3) combinatorial constraints is an extremely challenging problem that does not admit gradient-based methods (e.g., à la projected gradient descent for adversarial attacks) or polynomial-time algorithms. In practice, we impose a time limit to the MILP solver and modify the solver’s parameter settings to prioritize finding feasible solutions quickly over proving global optimality.
>
> 5. **Q-learning for continuous action spaces:** Q-learning has indeed been used for continuous action spaces, but standard Q-learning (or any existing modifications) cannot solve a policy that says: “Find me the best policy that solves an assignment problem (an NP-hard discrete optimization problem) at every timestep.” This is what our work achieves here, to integrate MIP solving into deep RL.
>
> 6. **Elaboration on challenges embedding a NN into a MIP:** Others have shown that neural networks can be embedded into a MIP, but our paper is the first to integrate **deep RL** (not just deep learning) with MIP solving to enable combinatorial action constraints. Please see our response to Weakness #4 for more detail on the technical challenges.
>
>
> ### References
> > Mate, et al. Field study in deploying restless multi-armed bandits: Assisting non-profits in improving maternal and child health. AAAI 2022
> >
> > Raman, Shi, and Fang. Global rewards in restless multi-armed bandits. NeurIPS 2024
> >
> > Whittle. Restless bandits: Activity allocation in a changing world. Journal of Applied Probability 1988
> > Weber and Weiss. On an index policy for restless bandits. Journal of Applied Probability 1990

---

> ### Comment · Reviewer_EMxA · 2024-11-30
> **Thank you for your response and follow-up about scalability**
>
> Thank you for your response which clarified some of my concerns. I would like to follow-up on the main motivation of the paper and the main challenge to be addressed which is scalability.
>
> One of the main motivations of the paper is to address combinatorial action spaces which bring an important scalability challenge into picture. I have a few follow-up questions for clarification regarding some of the bottlenecks which are also mentioned in the paper:
>
> 1. The paper proposes to embed the Q function into an MILP and then solve it to find the maximizing action (in principle). However, embedding (by linearizing large NNs, is this done automatically?) can result in a very large MILP which can quickly become intractable to solve (these are also arguably very hard problems though as mentioned by the authors). In practice these can take days to be solved even for small instances. Moreover, the algorithm has to solve an MILP **for each sampled state within each episode**. Given the number of samples usually required to train RL, this is huge. I am wondering how do you meaningfully reduce the complexity of the MILPs to be solved (which have among decision variables the huge combinatorial action space).
> The reported running time in this work for the training is about 1 to a few hours. You mention that you 'impose a time limit to the MILP solver and modify the solver’s parameter settings to prioritize finding feasible solutions quickly over proving global optimality'. So do you still keep the same large combinatorial action space and by just reducing the running time you are able to get a feasible action to the problem, how can you guarantee that you get such an action just by reducing the running time?
> The paper argues that considering a Q-network with an output of the size of the combinatorial action space is intractable, but now with the proposed approach the max over the Q-network is still intractable and it has to be solved a large number of times. The experiments provide some evidence but could you elaborate more on how you manage to obtain a better result with MILP with a very limited running time that would be better than any random feasible action?
>
> 2. The exploration strategy which is also discussed in section 4.2 is based on heuristics. I find it hard to get any meaningful estimates on actions that are never encountered for instance. If all of them are sampled then you also need exponentially many sampled actions which is intractable. How would the values learned for the Q function for specific actions inform or could be extrapolated to unseen actions even intuitively?  Otherwise there should be a cost to that in the approximation, does the approach in the paper involve indirectly reducing the number of actions sampled compared to the combinatorial size of the action set? Are you somehow exploiting some problem specific structure that drastically reduces the number of feasible actions in the combinatorial action space?
>
> 3. The proposed algorithm outputs a Q function. Does it mean that to find the action to be performed at a given state (i.e. the policy), you have solve an MILP (i.e. for each given state)?
>
> 4. Minor: The transition operator is valued in $[0,1]^J$ in Eq. 1. Is this a typo and it should be just [0,1]? What is the meaning of the Bellman equation you provide otherwise?
>
> I understand that this is a hard problem to solve and any meaningful progress is important. I would like to sense better how the central scalability challenge is meaningfully addressed here, could you elaborate more on this matter?

---

> > ### Author Response · Authors · 2024-12-01
> >
> > Thank you for your careful attention to our paper; we're glad to hear that our response clarified some of your concerns.
> >
> > As you highlight, this is indeed a hard problem to solve, and we appreciate your recognition that any meaningful progress is important.
> >
> > To address your follow-up questions:
> >
> > ## Question 1: Efficiently sampling states
> >
> > As we mention in line 291, embedding the neural network into the MILP can be done adding a linear $\mathcal{O}(DP)$ binary variables and constraints, where $D$ are the number of hidden layers and $P$ are the number of neurons per layer. In the experiments, we use $D=2$ layers and $P= \lbrace 32, 128 \rbrace $ hidden neurons, which we show is sufficiently expressive to capture the problem sizes we consider — despite the largest setting (with $J=100$ arms and $N=20$ workers) having up to $\approx 5.2 \times 10^{20}$ possible inputs.
> >
> > To embed the neural network, we have a custom translator from a trained PyTorch model to a mixed-integer linear problem in Gurobi. However, there are now automatic translators including from Gurobi which would facilitate this process even further (https://gurobi-machinelearning.readthedocs.io).
> >
> > Modern MILP solvers, such as Gurobi which we use, are designed to find optimal solutions as efficiently as possible. For example, Berthold [2013] shows that even when an optimal solution takes over 3,000 seconds to compute, a solution that achieves 90% of the same reward can be found within less than 800 seconds (Fig. 1 and 2).
> >
> > In all 12 of our experiment settings (Figure 3), clearly our SEQUOIA approach performs better than just selecting a random feasible action. Imposing a time limit for every MILP solve is used only as a safeguard, to avoid excessively long runs (rare in our experience). For all the problems we have considered, the solver finds a feasible solution very quickly, making early termination possible.
> >
> >
> > > Berthold (2013). Measuring the impact of primal heuristics. Operations Research Letters
> >
> >
> >
> > ## Question 2: Exploration strategy
> > As we discuss in Section 4.2, generating efficient and informative samples to train the Q-function was a priority for developing our method. Important to note is that to simply evaluate the immediate reward of a $(\mathbf{s},\mathbf{a})$ pair only requires computing the reward function, and does not require solving the MILP at all. We warm-start our Q-network with many of these such samples (lines 327-337).
> >
> > As you correctly inquire, we do have problem structure that we can also helpfully leverage. We discuss this in lines 349-353: one particularly useful property of restless bandits is that the impact of an action on each arm is decoupled, as state transitions are defined independently per-arm. Thus, the transition dynamics are less complex to learn, and we can even simulate valid state transitions even for infeasible actions (e.g., actions that exceed a budget constraint).
> >
> > Altogether, these conditions combined enable us to tractably explore the large action set. And, as shown in our experiments, we are able to achieve strong performance within reasonable time limits.
> >
> >
> > ## Question 3: Going from Q-function to action
> > Correct, Algorithm 1 is the process for training the Q-function. At inference time, to find the action to be performed at a given state, we perform a single solve of the MILP to find the best action $\mathbf{a}$ from the current state $\mathbf{s}$.
> >
> >
> > ## Question 4: Transition operator
> > The joint transition probability $P^\times$ is indeed over $[0, 1]^J$ (no typo). This is the joint transition probability for all $J$ arms. The state for each arm $j \in [J]$ is in the range $[0, 1]$.
> >
> > The Bellman equation in eq. (1) is over the joint state space — note that we use the vector notation $\mathbf{s} \in \mathcal{S}^\times$, where $\mathbf{s}$ is a vector of length $J$, representing the current state of each arm $j$.
> >
> >
> > ———
> >
> > We hope that our responses sufficiently clarify your remaining questions! If so, we would appreciate your updating your score accordingly.

---

> ### Comment · Reviewer_EMxA · 2024-12-01
> **Thank you**
>
> Thank you for your response, I am increasing my score given the importance of the problem and its applications (with the settings highlighted) as well as the effort of the paper to go beyond non-combinatorial action spaces in practice. The paper builds on prior work to embed a neural network into an MILP and uses some heuristics to address the issue of scaling for combinatorial action spaces.  I still believe there are a number of issues to be addressed properly in terms of scaling as this is one of the main goals of the paper: solving an MILP for each state sampled seems quite restrictive given the usual number of samples one needs in RL for instance although the method improves over the baselines presented on the instances presented, exploration is handled with some heuristic sampling that does not show the limitation of the proposed method in harder instances, I believe the justifications provided in the rebuttal need to enhance the paper since scalability is the key challenge, the Q-network used has 2 hidden layers of size 32 each which seems quite small to handle large problems with combinatorial action spaces and training a neural network with 128 neurons takes an order of magnitude longer in terms of running time as mentioned in appendix G. Given that the paper is fully practical (and no guarantees are provided), I would expect a more extensive and comprehensive experimental study to support the proposed method (on more instances, larger ones, investigating sensitivity to the parameters ...), its scalability and show its limitations.

---

### Official Review · Reviewer_oEWh · 2024-11-09

**Soundness:** 3
**Presentation:** 3
**Contribution:** 2
**Rating:** 6
**Confidence:** 2

**Summary:**

This paper introduces a novel reinforcement learning framework for a challenging setting known as combinatorial restless multi-armed bandits (CoRMAB). In these problems, the vast combinatorial action space presents a key bottleneck, especially for real-world applications like public health. The authors address this by proposing SEQUOIA, a method that combines a Q-network with mixed-integer linear programming (MILP) solvers. Four distinct constraint types (such as capacity and matching constraints) are applied to CoRMAB instances to explore the method's performance. Experimental results show that SEQUOIA significantly outperforms existing baselines across these settings.

**Strengths:**

- The paper highlights meaningful applications, particularly in public health, where combinatorial decision-making is crucial. By focusing on real-world-inspired constraints, the paper emphasizes the practical utility of SEQUOIA.
- The experimental results suggest that SEQUOIA has a strong advantage over other methods, particularly in scenarios that require both sequential planning and combinatorial action selection. This shows potential for the method to be impactful in complex decision-making tasks.

**Weaknesses:**

- The four specific CoRMAB instantiations appear somewhat interdependent. For instance, the first instance involving multiple interventions seems to implicitly contain elements of the second (path constraints) and third (capacity constraints). This overlap could obscure the unique contributions of each instance, and the presentation of these distinctions would benefit from clarification. Additionally, a more precise formulation of the optimization goals and constraints in each problem setting could strengthen the paper.
- Although the SEQUOIA framework is innovative in combining Q-networks with MILP, the method lacks theoretical guarantees, which may reduce its general appeal in theoretical RL circles. The paper leans toward practical applications without rigorously addressing theoretical underpinnings. Given that the CoRMAB problems are motivated by real-world scenarios, it would be beneficial for the authors to demonstrate how SEQUOIA could operate on actual datasets or real-world instances.
- The method's training demands significant computational resources, as evidenced by Table 3, where training times extend to hours. For online applications, this can be a prohibitive factor. The paper would benefit from a discussion on optimizing computational efficiency or alternative approaches to reduce overhead.

**Questions:**

- Since SEQUOIA’s primary application appears to be in public health, there are concerns about performance during the initial training phase. If the network’s early-stage predictions are suboptimal, this could lead to unacceptable decisions in real-world use. How do the authors envision mitigating this issue in practice, particularly in high-stakes domains like public health where early errors could have critical impacts?

---

> ### Author Response · Authors · 2024-11-13
>
> Thank you for your review and suggestions! We're happy that you find the applications meaningful and that our method is practical and has potential for impact.
>
> We will update the paper to reflect the following response.
>
> ## Response to your question (performance during initial training):
> Thank you for raising this concern about deployment in high-stakes environments. As discussed in Section 2, the coRMAB setting assumes that the environment is known, i.e., the transition probabilities and reward functions have already been derived. Offline planning is the standard assumption for restless bandits, and has been deployed for important real-world settings including public health in India [1] and clinical trial design [2]. In [1], they build the offline simulator (of the probability of a patient becoming engaged in a health program following a messaging intervention) using domain experts and historical data. As such, training the Q-network using Algorithm 1 is done in simulation, offline. Once satisfactory performance is achieved, the policy can be deployed in the real world.
>
> This offline planning approach is discussed in Section 4.1 “Learning in the Real World” of [3], as a way of addressing online RL’s extensive data needs. Should SEQUOIA be applied to an environment with unknown dynamics, we recommend building a model of the environment dynamics first as discussed in [4], and then applying our Algorithm 1 before finally deploying the Q-network.
>
> ## Response to the Weaknesses
>
> 1. **The four coRMAB instantiations:** We provide precise formulations of the goals and constraints of all 4 coRMAB problem settings we introduce in full mathematical detail in Appendix C. However, we disagree that these instantiations are all interdependent. For example, for the capacity-constrained setting, each worker $j$ can act on multiple arms (up to their budget $b_j$), but in the schedule-constrained setting, each worker can only act on a single arm — and based on incompatible availability, some workers may not be assigned at all. Whereas for the path-constrained setting, the “number of workers” is not a fixed number $N$, but rather corresponds to the total length of a feasible path. Separately, in the multiple interventions setting, the impact of multiple workers acting on the same arm is cumulative; that is not the case in any of the other settings (i.e., multiple workers acting on arm $j$ is no better than one worker acting on arm $j$).
>
> 2. **Theoretical guarantees:** The vast majority of deep RL approaches focus on empirical performance, rather than on theoretical guarantees. For example, DQN (the deep RL algorithm we use) was introduced in 2015, but the first attempt for a theoretical analysis of DQN was not done until 2020 [5] — and this paper has to make extremely simplifying assumptions to get convergence guarantees, such as assuming the data-gathering policy has good coverage (thus requiring minimal exploration) and that the trained neural network is sparse (i.e., mostly 0s). Other key deep RL algorithms, such as DDPG, PPO, or RAINBOW, do not come with theoretical guarantees.
>
> As you highlight, our focus is on practical restless combinatorial bandit problems. We have therefore proposed realistic problems and realistic generation schemes.
>
> 3. **Computational runtime:** As we mentioned above, SEQUOIA is an offline planning approach. As such, a few hours of training can be seen as a reasonable upfront cost before deployment.
>
> > [1] Mate, Madaan, Taneja, et al. “Field study in deploying restless multi- armed bandits: Assisting non-profits in improving maternal and child health.” In AAAI 2022.
> >
> > [2] Villar, Bowden, Wason. “Multi-armed bandit models for the optimal design of clinical trials: benefits and challenges.” Statistical Science: A Review Journal of the Institute of Mathematical Statistics 2015.
> >
> > [3] Whittlestone, Arulkumaran, and Crosby. "The societal implications of deep reinforcement learning." JAIR 2021.
> >
> > [4] Moerland, et al. "Model-based reinforcement learning: A survey." Foundations and Trends in Machine Learning 2023.
> >
> > [5] Fan, et al. "A theoretical analysis of deep Q-learning." Learning for dynamics and control. PMLR, 2020.

---

> > ### Comment · Reviewer_oEWh · 2024-12-02
> >
> > Thank you for your responses. I will reconsider my score.

---

### Official Review · Reviewer_1mgZ · 2024-12-03

**Soundness:** 4
**Presentation:** 4
**Contribution:** 4
**Rating:** 6
**Confidence:** 3

**Summary:**

This paper considers the CoRMAB problem, in which there are complex combinatorial arm structures to be learned. Such complexity often arises in many real-world applications such as public health etc. The problem, as far as I can see, is kind of like an intermediate area between traditional bandits where each arm is independent, and that of the more general Markov Decision process where any transition might happen. This paper, however, focuses on the more tractable scenario where some sort of information is known beforehand about the arm dependence structure. In particular, the paper considers four specific scenarios, including multiple interventions, bipartite matching, capacity constrained, and path planning. The paper proposed SEQUOIA, which applies Q-network with mixed-integer linear programming to solve the problem. Experiments demonstrate the effectiveness and efficiency of the proposed algorithm.

**Strengths:**

The paper studied a very novel and niche problem that is more tractable than RL but also more practically relevant in real-world bandit applications. I think it's important to have deeper investigations on such problems to directly applying SOTA RL algorithms. One novelty is that the paper combines RL with MILP to solve the problem more effectively.

**Weaknesses:**

Although the paper studied very interesting bandit problems, but the problem is solved via RL plus MILP. I was curious why not directly apply SOTA RL algorithms? How does that compare to the SEQUOIA proposed in this paper? I think one weakness of the paper is that it didn't compare with more advanced baselines like certain RL algorithms.

Another major weakness of the paper is that it doesn't have theoretical analysis on the algorithm performance, which is very critical for bandit papers. I would hope the authors provide regret bounds for each of the four cases studied in the paper.

**Questions:**

How does the algorithm in this paper compare to SOTA RL algorithms?

How to derive theoretical analysis?

---

> ### Author Response · Authors · 2024-12-03
>
> Thank you for your review! We are glad to hear you find our problem interesting, novel, and relevant for real-world applications.
>
> ## Question: Why no regret bounds, unlike traditional bandit papers?
>
> We would like to clarify the distinction between restless bandits (RMABs) and traditional multi-armed bandits. Our paper considers RMABs, which are typically a planning problem that assume known transition dynamics. The challenge for RMABs is tractably planning over this large, budgeted action space. As you point out, RMABs are a form of Markov decision processes, where each arm of the RMAB is an MDP.
>
> On the other hand, traditional bandits are *learning* algorithms thus study theoretical guarantees, measured in terms of regret. Regret would be a useful metric to compare if we are considering a learning setting with unknown transition dynamics. However, we focus instead on planning with known transition dynamics, where the challenge is overcoming computational complexity, not learning unknown dynamics.
>
> For other convergence bounds, note that the vast majority of deep RL approaches focus on empirical performance, not on theoretical guarantees. For example, DQN (the deep RL algorithm we use) was introduced in 2015, but the first attempt for a theoretical analysis of DQN was not done until 2020 [Fan et al. 2020] — and this paper makes extremely simplifying assumptions, such as assuming the data-gathering policy has good coverage (thus requiring minimal exploration) and that the trained neural network is sparse (i.e., mostly 0s).
>
> As mentioned in line 487, Tkachuk et al. (2023) recently offered a theoretical analysis of RL with combinatorial actions under some linearity assumptions. While not applicable to our setting, it could serve as a starting point for further theory in the future.
>
> > Fan, et al. "A theoretical analysis of deep Q-learning." Learning for dynamics and control. PMLR, 2020.
> >
> > Tkachuk et al. "Efficient planning in combinatorial action spaces with applications to cooperative multi-agent reinforcement learning." AISTATS 2023.
>
>
> ## Question: Comparison to SOTA RL algorithms?
>
> As emphasized in the paper, our action space is both combinatorial and subject to hard constraints. Naively “expanding” all possible actions results in an extremely large discrete action space. In Figure 2, we illustrate how this combinatorial explosion makes the use of existing RL methods impossible.
>
> For example, we show how a “Standard DQN” in the top-left corner would require a Q-network with as many outputs as actions. For our largest setting in the experiments ($J=100$ arms and $N=20$ workers), the number of possible actions is $\approx 5.3 \times 10^{20}$. In addition, not all binary vectors are feasible actions, as they may violate constraints (e.g., path constraints, assignment constraints, budget constraints, etc.). This limits the applicability of SOTA RL methods. Part of our contribution is to show that using MILP within a value-based RL approach can help address both the combinatorial explosion and the presence of hard constraints.
>
> ——
>
> Thank you again for your review of our paper; we hope our response addresses your concerns and helps to contextualize our contributions!

---

### Meta-Review · Area_Chair_XPR6 · 2024-12-21

**Metareview:**

This paper studies reinforcement learning (RL) with combinatorial action spaces, specifically within the novel coRMAB framework for coupled restless bandits. It presents significant empirical results using SEQUOIA, an RL algorithm that integrates mixed-integer programming with deep Q-networks to optimize long-term rewards under combinatorial constraints. The reviewers appreciated the practical relevance, novelty of the problem setting, and the effectiveness of the proposed method, as highlighted by strong experimental results and detailed rebuttals addressing initial concerns. Although one rejecting reviewer emphasized the lack of large-scale experiments and theoretical guarantees, this critique is less applicable as the paper's primary focus is theoretical contributions and computational techniques for RL. Overall, the paper's advancements in handling large, combinatorially structured action spaces in RL warrant acceptance for their theoretical and practical impact.

**Additional Comments On Reviewer Discussion:**

During the rebuttal period, the reviewers raised several key points. Reviewer 1mgZ emphasized the lack of theoretical guarantees like regret bounds and comparisons to state-of-the-art (SOTA) RL algorithms; the authors clarified the distinction between planning and learning settings, explaining why regret does not apply and why comparisons to SOTA RL methods were infeasible given the combinatorial action space. Reviewer oEWh raised concerns about scalability and overlapping problem settings; the authors provided details on their MILP-solving strategy, clarified the distinctiveness of the scenarios, and highlighted practical performance benefits. Reviewer EMxA questioned scalability with large neural networks and MILPs; the authors explained their heuristic methods, MILP optimizations, and problem-specific structure exploitation. Reviewer ya5G suggested broader experimental studies and comparisons to domain-specific solutions; the authors acknowledged these points but noted the lack of suitable baselines for sequential, combinatorial action problems. Reviewer 9UJj requested more clarity on specific formulations and noted the lack of real-world datasets; the authors updated explanations and acknowledged the data limitation. Each concern was thoughtfully addressed, and the paper's focus on theoretical contributions and computational techniques overcomes its limitations, supporting an acceptance recommendation.

---

### Decision · Program_Chairs · 2025-01-22

Accept (Poster)